# THE OVERSMOOTHING FALLACY: A MISGUIDED NARRATIVE IN GNN RESEARCH

## ABSTRACT

Oversmoothing has been recognized as a main obstacle to building deep Graph Neural Networks (GNNs), limiting the performance. This paper argues that the influence of oversmoothing has been overstated and advocates for a further exploration of deep GNN architectures. Given the three core operations of GNNs, aggregation, linear transformation, and non-linear activation, we show that prior studies have mistakenly confused oversmoothing with the vanishing gradient, caused by transformation and activation rather than aggregation. Our finding challenges prior beliefs that oversmoothing in GNNs is dominated by the GNN-specific structure of aggregation. Furthermore, we demonstrate that classical solutions such as skip connections and normalization enable the successful stacking of deep GNN layers without performance degradation. Our results clarify misconceptions about oversmoothing and highlight the untapped potential of deep GNNs.

## 1 INTRODUCTION

Motivated by the success of deeper neural networks in various domains, many studies have sought to increase the depth of Graph Neural Networks (GNNs) to achieve better performance. However, these attempts often result in performance degradation rather than improvement, primarily attributed to a phenomenon known as oversmoothing, a situation in which each node embedding converges to a non-informative point as the number of layers increases (Li et al., 2018).

The concept of oversmoothing was introduced by Li et al. (2018), quickly gaining acceptance within the community due to the intuitive idea that aggregation steps inherently cause embeddings of neighboring nodes to become similar. Early theoretical studies confirmed the asymptotic emergence of oversmoothing. Subsequent works, including those by Oono & Suzuki (2020) and Wu et al. (2023a), extended this perspective by demonstrating that oversmoothing measures decrease exponentially even for relatively shallow GNNs. Following this, numerous methodological innovations, including novel model architectures, regularization strategies, and normalization methods, were proposed to better analyze and mitigate oversmoothing (Chen et al., 2020; Fang et al., 2023; Lu et al., 2024; Park et al., 2024; Rong et al., 2020; Su et al., 2024; Zhao & Akoglu, 2020; Zhou et al., 2021a).

This paper argues that there exists a common misunderstanding of oversmoothing in the community, and the actual influence of oversmoothing induced from a unique property of GNN is marginal. First, we observe that despite the initial identification of two distinct definitions of oversmoothing, subsequent studies typically adopted only one, creating confusion and hindering a clear theoretical understanding. Moreover, many previous studies overly emphasize the role of the aggregation step in oversmoothing, among the various components of GNNs. To address this gap, we systematically analyze the contributions of each of the three core components of GNN layers: i) aggregation, ii) linear transformation, and iii) nonlinear activation, to the oversmoothing.

Our analysis quantitatively demonstrates that, contrary to popular belief, aggregation alone has a minimal impact on oversmoothing. Instead, our findings reveal that the nonlinear activation and linear transformation steps are responsible for the *zero-collapsing* phenomenon, whereby node embeddings converge toward the zero vector. Therefore, we conjecture that many previous studies have mistakenly confused zero-collapsing with oversmoothing. Through additional experiments, we demonstrate that when oversmoothing occurs (distinct from zero-collapsing), GNNs can still

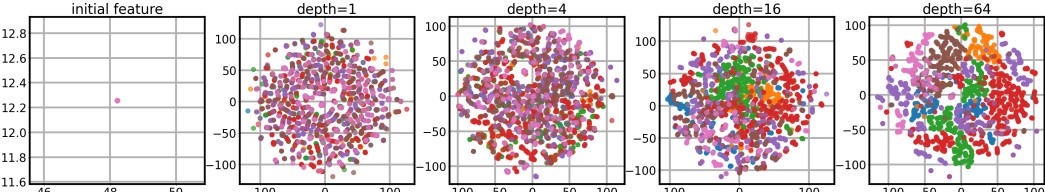

Figure 1: Changes in node embeddings across GCN layers, where all node features are initialized to the same value (thus initially oversmoothed), visualized via t-SNE (Van der Maaten & Hinton, 2008). The result implies that *a GCN can recover from oversmoothing even after it has occurred*.

effectively learn and mitigate its effects. Figure 1 illustrates how input node features, initialized identically across all nodes, evolve across layers when a model is properly trained.

Zero-collapsing has long been observed in traditional multilayer perceptrons (MLPs) and recurrent neural networks, and it is widely recognized as a leading cause of vanishing gradients. Leveraging this insight, we apply well-established solutions for mitigating zero-collapsing, the residual connections and the batch normalization, to GNN architectures. Remarkably, we show that by properly integrating these two simple yet effective strategies, it is feasible to train extremely deep GNNs, successfully scaling up to 1,024 layers. Finally, we reflect on why the oversmoothing research community has treated the vanishing gradient, a classic deep learning problem, as a unique property of GNNs and discuss how this perception has limited research on deep GNNs.

We summarize the key messages of our work as follows:

- We clarify the common misconception that oversmoothing, typically understood as embeddings becoming similar, only reflects one of two fundamentally distinct phenomena.

- We demonstrate that many prior studies have incorrectly labeled zero-collapsing as oversmoothing and clarify that the aggregation step in GNNs has a marginal effect on oversmoothing.

- We show that the celebrated graph convolutional network can effectively overcome oversmoothing, provided that zero-collapsing is addressed separately.

- We empirically verify that standard techniques used to mitigate zero-collapsing in neural networks, such as residual connections and batch normalization, remain efficient solutions.

- Finally, we discuss why the current research community is missing this seemingly evident conclusion and provide historical context and potential reasons.

**Remark.** Our work is *not in contradiction* to the previous theoretical work on oversmoothing. Several theoretical works rigorously establish that oversmoothing arises in *linearized or asymptotic regimes*. For instance, Keriven (2022) show oversmoothing as a consequence of stochastic matrix convergence. More recently, Roth & Liebig (2024) demonstrate that the spectral properties of the aggregation operator force node representations into a low-dimensional invariant subspace regardless of feature transformations with linearized models. These results provide valuable insight into the limiting behavior of message-passing networks, but they largely characterize *asymptotic convergence* rather than explaining the degradation observed at practical depths. In contrast, our work emphasizes that in finite-depth, trainable GNNs, the dominant obstacle is zero-collapsing and vanishing gradients from transformations and activations, which can be mitigated with standard deep-learning strategies. *We therefore view these asymptotic analyses as complementary*: they describe the ultimate fate of infinitely deep propagation, while our focus is on the optimization bottlenecks that matter in practice.

## 2 BACKGROUND

The term "oversmoothing" has permeated the graph neural network (GNN) community, often invoked to explain why deeper GNNs fail to yield improved performance. In this section, we introduce

a general framework of GNN with two representative examples (Section 2.1), a formal description of oversmoothing (Section 2.2), and related works proposed to solve the oversmoothing (Appendix A).

## 2.1 GENTLE INTRODUCTION TO GRAPH NEURAL NETWORKS

We consider an undirected graph $\mathcal{G} = (\mathcal{V}, \mathcal{E})$, where $\mathcal{V}$ is a set of $N \in \mathbb{N}$ nodes and $\mathcal{E} \subseteq \mathcal{V} \times \mathcal{V}$ is a set of edges, with additionally observed $d$-dimensional node feature $\mathbf{X} \in \mathbb{R}^{N \times d}$. We denote the $d$-dimensional feature of node $i$ as $\mathbf{x}_i \in \mathbb{R}^d$. The connectivity between nodes in an undirected and unweighted graph can also be represented through a binary symmetric adjacency matrix $\mathbf{A} \in \{0,1\}^{N \times N}$, where an edge between node $i$ and $j$ is represented as an entry $\mathbf{A}_{ij} = \mathbf{A}_{ji} = 1$.

In this work, we consider two variants of GNNs, graph convolutional network (GCN) (Kipf & Welling, 2017) and graph attention network (GAT) (Veličković et al., 2018), since these two models are considered as "the target model" that needs to be fixed in many previous research on over-smoothing. Both GCN and GAT transform the input feature $\mathbf{X}$ through $L$ layers. At each layer $\ell \in [0, L-1]$, input node feature $\mathbf{X}^{(\ell)}$ is updated as

$$\mathbf{X}^{(\ell+1)} = \sigma(\hat{\mathbf{A}}\mathbf{X}^{(\ell)}\mathbf{W}^{(\ell)}), \tag{1}$$

where $\mathbf{X}^{(0)} := \mathbf{X}$, $\sigma(\cdot)$ is a non-linear activation function, and $\mathbf{W}^{(\ell)}$ is a learnable weight matrix at layer $\ell$. The matrix $\hat{\mathbf{A}}$ serves as the aggregation operator, and its structure differs significantly between GCN and GAT. In GCN, $\hat{\mathbf{A}}$ is a fixed, symmetrically normalized adjacency matrix defined as $\hat{\mathbf{A}}_{\mathrm{GCN}} = \tilde{\mathbf{D}}^{-\frac{1}{2}}\tilde{\mathbf{A}}\tilde{\mathbf{D}}^{-\frac{1}{2}}$ where $\tilde{\mathbf{A}} = \mathbf{A} + \mathbf{I}$ is an adjacency matrix with self-loops, $\tilde{\mathbf{D}}$ is a degree matrix with diagonal entries $(\tilde{\mathbf{D}})_{ii} = d_i + 1$, and $d_i = \sum_{j=1}^{N} \mathbf{A}_{ij}$ is the node degree. In contrast, GAT constructs $\hat{\mathbf{A}}$ dynamically using a learnable attention mechanism. The resulting matrix $\hat{\mathbf{A}}_{\mathrm{GAT}}$ is right-stochastic, satisfying $\sum_j (\hat{\mathbf{A}}_{\mathrm{GAT}})_{ij} = 1$ for all $i$.

A single layer of GCN and GAT can be factorized further into three sequential steps: aggregation, transformation, and activation. Formally, these steps are expressed as

$$\hat{\mathbf{X}}^{(\ell+1)} = \hat{\mathbf{A}}\mathbf{X}^{(\ell)} \text{ (aggregation)}, \tag{2}$$

$$\tilde{\mathbf{X}}^{(\ell+1)} = \hat{\mathbf{X}}^{(\ell+1)}\mathbf{W}^{(\ell)} \text{ (transformation)}, \tag{3}$$

$$\mathbf{X}^{(\ell+1)} = \sigma(\tilde{\mathbf{X}}^{(\ell+1)}) \text{ (activation)}. \tag{4}$$

We consider Rectified Linear Unit (ReLU) as the activation function, which has been widely used in most GNN architectures (Kipf & Welling, 2017; Veličković et al., 2018; Wu et al., 2023a).

## 2.2 A BRIEF REVIEW OF STUDIES DEMONSTRATING OVERSMOOTHING

In the literature, we identify two distinct definitions of oversmoothing, each specific to GCNs and GATs, respectively. The following two propositions clarify the distinction between the oversmoothing in GCNs and GATs.

**Proposition 1** (degree-scaled embedding convergence (Cai & Wang, 2020; Li et al., 2018; Oono & Suzuki, 2020)). *In GCNs, there exist $\mathbf{c} \in \mathbb{R}^d$ such that $\lim_{\ell \to \infty} \mathbf{x}_i^{(\ell)} = \sqrt{d_i + 1}\mathbf{c}$ for all nodes $i \in \mathcal{V}$*

**Proposition 2** (uniform embedding convergence (Keriven, 2022; Thorpe et al., 2022; Wu et al., 2023a)). *In GATs, there exist $\mathbf{c} \in \mathbb{R}^d$ such that $\lim_{\ell \to \infty} \mathbf{x}_i^{(\ell)} = \mathbf{c}$ for all nodes $i \in \mathcal{V}$*

These propositions imply that unless all node degrees are equal, i.e., $d_i = d$ for all $i$, or embeddings converge to the zero vector, i.e., $\lim_{\ell \to \infty} \mathbf{x}_i^{(\ell)} = [0, 0, \cdots, 0]^\top$, GCNs and GATs exhibit distinct behaviors in the asymptotic regime.

Given the two propositions, we briefly summarize prior works that provide proofs of oversmoothing. Earlier studies paid attention to the asymptotic behavior of the aggregation step, demonstrating that oversmoothing occurs when the aggregation process of GNNs is infinitely repeated. Li et al. (2018) show that the aggregation process of a GCN is a special form of Laplacian smoothing, resulting in the degree-scaled embedding convergence. Keriven (2022) demonstrate that the repeated aggregation process with the right stochastic matrix results in the uniform embedding convergence, using the

ergodic theorem. Chamberlain et al. (2021) interpret GATs as a discretization of the heat diffusion equation, implying that the embedding reaches equilibrium. Based on this intuition, Thorpe et al. (2022) provide a theoretical analysis showing that the outputs of GATs asymptotically converge to a fixed value independent of the initial inputs. However, such asymptotic analyses do not explain why performance degradation occurs in GNNs within the non-asymptotic regime, that is, when GNNs have only a small number of layers.

Recent studies (Cai & Wang, 2020; Oono & Suzuki, 2020; Wu et al., 2023a), providing a theoretical analysis that the oversmoothing occurs at an exponential rate, seem to answer the question. To quantify the speed of oversmoothing, these studies first define a node similarity measure $\mu : \mathbb{R}^{N \times d} \to \mathbb{R}_{\geq 0}$ that approach zero under conditions satisfying either Proposition 1, Proposition 2, or both. With the proposed measure, the exponential oversmoothing can be shown as exponential decay of the measure, such that

$$\mu(\mathbf{X}^{(\ell)}) \leq Cq^\ell \text{ for } \ell = 1, \cdots, L - 1, \tag{5}$$

for some constant $C > 0$, and $0 < q < 1$. In Appendix B, we summarize four representative oversmoothing measures, $\mu_{\text{oono}}$, $\mu_{\text{cai}}$, $\mu_{\text{wu}}$, and $\mu_{\text{rusch}}$ from Oono & Suzuki (2020); Cai & Wang (2020); Wu et al. (2023a); Rusch et al. (2023a), respectively, and the exponential decay rate of embeddings with each measure. It is worth noting that Oono & Suzuki (2020); Cai & Wang (2020) prove the decay rate for each individual step described in Equations (2) to (4).

Based on these analysis, some studies attempt to overcome the oversmoothing when stacking multiple layers. We summarizes these additional techniques in Appendix A.

## 3 AGGREGATION UNFAIRLY BLAMED FOR OVERSMOOTHING

In this section, we argue that the property of the aggregation step in Equation (2) has been misunderstood under the oversmoothing perspective. Consequently, the true influence of aggregation has been overestimated, obscuring the real cause of performance degradation. To do so, we first describe a common misconception about oversmoothing and a potential reason that causes the misunderstanding. To understand why the misunderstanding happens, we then precisely measure the influence of individual steps of Equations (2) to (4) in oversmoothing. Through the analysis of the results, we find that the aggregation step has a marginal influence.

### 3.1 THE FORGOTTEN TWO TAILS OF OVERSMOOTHING

*A common **misconception** is that oversmoothing is simply the phenomenon where node embeddings become increasingly similar as the number of layers grows (Fang et al., 2023; Li et al., 2019; Lu et al., 2024; Rusch et al., 2023a;b; Song et al., 2023; Thorpe et al., 2022; Wu et al., 2023a; Zhang et al., 2022; Zhou et al., 2021b).* However, interpreting oversmoothing solely from this perspective can be misleading, dangerous, and inappropriate, as there are fundamentally different definitions of oversmoothing as we have shown in Propositions 1 and 2, and the misconception only represents Proposition 2.

We investigate the reasons why researchers became confused about oversmoothing. One interesting observation is the contradictory result presented by Rusch et al. (2023a), who empirically show embeddings of all nodes converging to a single point in a GCN. This finding contradicts the theoretical result stated in Proposition 1, which predicts convergence of embeddings scaled by node degrees. Specifically, they proposed a smoothing measure $\mu_{\text{rusch}}$ (Equation (A7)) that approaches zero when only the condition of Proposition 2 is satisfied, and found that the GCN embeddings also approach zero according to this measure as shown in Figure 2. Unless the graph is regular, such a result conflicts with the theoretical analysis. What causes this discrepancy? In the following section, we analyze this issue in detail.

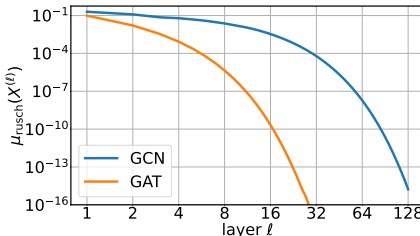

Figure 2: Similarity between nodes across layers, reported in Rusch et al. (2023a).

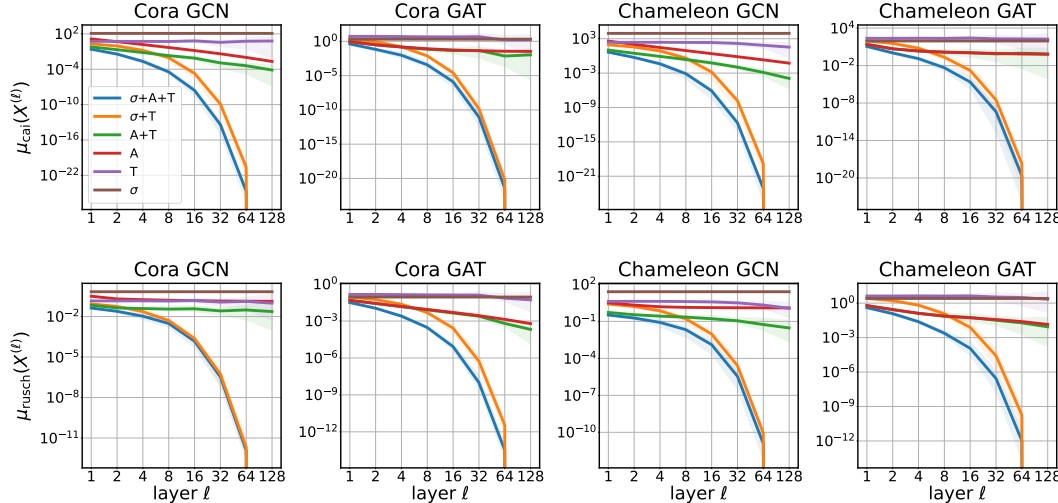

Figure 3: Oversmoothing levels measured by three metrics across 128-layer GCN and GAT models evaluated on two datasets: Cora and Chameleon. For each model, we compare four variations based on the combination of aggregation (A), transformation (T), and activation ($\sigma$) steps. For example, A+T refers to a model with only aggregation and transformation steps. *The oversmoothing is observable without the aggregation step.*

## 3.2 WE SHOULD DISTINGUISH BETWEEN OVERSMOOTHING AND ZERO COLLAPSING

We demonstrate that the exponential oversmoothing is not unique to GNNs but also occurs similarly in MLPs. Our analysis shows that oversmoothing induced solely by aggregation (Equation (2)) is negligible. Instead, the other two steps, transformation (Equation (3)) and activation (Equation (4)), are the primary contributors responsible for the exponential oversmoothing effect.

None of the previous studies report the influence of individual steps (Equations (2) to (4)) on the oversmoothing empirically. To quantify the independent influence of individual steps, we measure and compare the changes in node similarity using four variants of the GNNs: the original model ($\sigma$+A+T), the model with activation and transformation ($\sigma$+T), the model with aggregation and transformation (A+T), the model with aggregation-only (A), the model with transformation-only (T), and the model with activation only ($\sigma$). For example, the model with activation and transformation effectively becomes an MLP by excluding the aggregation step. The model with aggregation only can be seen as the SGC (Wu et al., 2019). We compute two node similarity measures, $\mu_{\texttt{cai}}(\mathbf{X}^{(\ell)})$ and $\mu_{\texttt{rusch}}(\mathbf{X}^{(\ell)})$, for $\ell = 1, 2, 4, \cdots, 128$ from the 128-layer GCN and GAT models on Cora (Sen et al., 2008) and Chameleon (Platonov et al., 2023; Rozemberczki et al., 2021) datasets.

Figure 3 shows the changes of the similarity measure over the layers. Regardless of the dataset and the node similarity measure, exponential oversmoothing is observed for the models with the activation step (i.e., $\sigma$+A+T, $\sigma$+T). The activation with transformation exhibits a decay rate similar to the original model, whereas the models excluding activation (i.e., A+T, A) show a significantly lower decay rate. Although a mild level of oversmoothing is observed with the aggregation-only model, the decay rate is much lower than that of models including activation. We present additional results on other datasets in Appendix D.1, where the same trend can be observed.

From these results, we conclude that *the influence of the aggregation step is marginal* and the oversmoothing is primarily caused by the combination of transformation and activation steps. Then why do transformation and activation steps cause oversmoothing? Note that $\sigma$+T is equal to the MLP architecture, and we already have a well-known case where the oversmoothing occurs with MLPs: the zero-collapsing, whereby node embeddings converge toward the zero vector (Glorot & Bengio, 2010).[1]

---

[1] Neuron saturation is used to describe a similar but different behavior with the sigmoid and hyperbolic tangent activations. We use 'zero-collapsing' to distinguish the specific scenario where neuron activations collapse towards zero, commonly observed with ReLU-like activation functions.

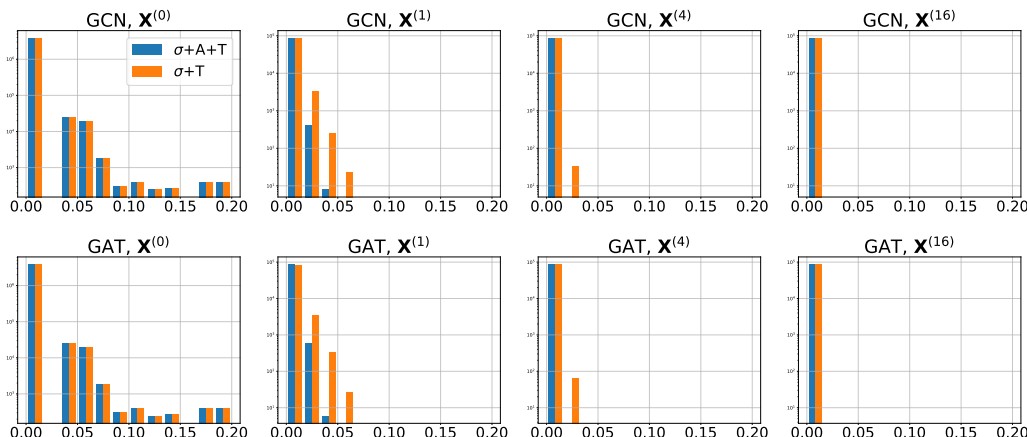

Figure 4: Histogram of node embedding values $\mathbf{X}^{(\ell)}$ from four selected layers. *All features are collapsed to zero within the first few layers.*

Figure 4 illustrates the distribution of node embedding values over the four selected layers of GCN and GAT. From the figure, one can easily observe that all features are collapsed to zero after the first few layers. Since the initial features are collapsed to zero in MLPs without proper normalization and residual connections, exponential oversmoothing is not a problem unique to GNNs. In MLPs, this phenomenon has been rather understood as a source of vanishing gradients, for which researchers have developed many solutions. Will these classical solutions, such as the residual connection and batch normalization, also work with GNNs? We answer the question in Section 4.1.

Note that we initialize the parameters using Glorot (Xavier) initialization (Glorot & Bengio, 2010) for the experiment. Parameter initialization has a significant impact on the experimental outcomes described above. The He initialization (He et al., 2015) is known to alleviate such issues effectively, whereas Glorot initialization is generally not considered optimal when paired with ReLU activation functions. One puzzling aspect we notice is that official implementations of GNNs provided by several libraries, such as Pytorch Geometric (Fey & Lenssen, 2019) and Chainer (Akiba et al., 2017; Tokui et al., 2015; 2019), adopt Glorot initialization by default, despite ReLU being the most popular activation function in GNNs. We find that numerous studies (Chen et al., 2020; Oono & Suzuki, 2020; Park et al., 2024; Rong et al., 2020; Rusch et al., 2022; 2023a;b; Zhao & Akoglu, 2020; Zhou et al., 2021a) also commonly adopt Glorot initialization in their official implementations. To show how the results change with different initialization choices, we provide the results using He initialization in Appendix D.5.

**In response to the question raised by the contradictory results discussed in Section 3.1**, we believe the confusion arises because the target embedding vector **c** in Propositions 1 and 2 includes the zero vector as a special case. Consequently, many previous studies incorrectly interpreted convergence to the zero vector as evidence of oversmoothing. However, our experiments suggest that convergence to the zero vector is not a genuine result of oversmoothing; rather, it occurs due to a zero-collapsing phenomenon. This subtle yet crucial distinction has led researchers to mistakenly attribute zero-collapsing to oversmoothing.

### 3.3 DEBUNKING THE MYTH: OVERSMOOTHING CAUSES VANISHING GRADIENT

As an interesting side experiment, we challenge the common misunderstanding that oversmoothing causes vanishing gradients and thus complicates optimization (Li et al., 2019; Rong et al., 2020). We provide a counterexample demonstrating that oversmoothing is not the primary cause of optimization failure. To explicitly reproduce the oversmoothing situation, one can replace input node feature $\mathbf{X}_i$ with an arbitrary fixed vector **c** for all nodes $i$ without manipulating the target labels $y_i$. Imagine a standard MLP without utilizing graph structure, the model cannot be trained properly under this configuration since the gradients have the same direction for any input $\mathbf{X}_i$. In the case of GATs, we can also expect that the node embeddings remain unchanged since the aggregation with

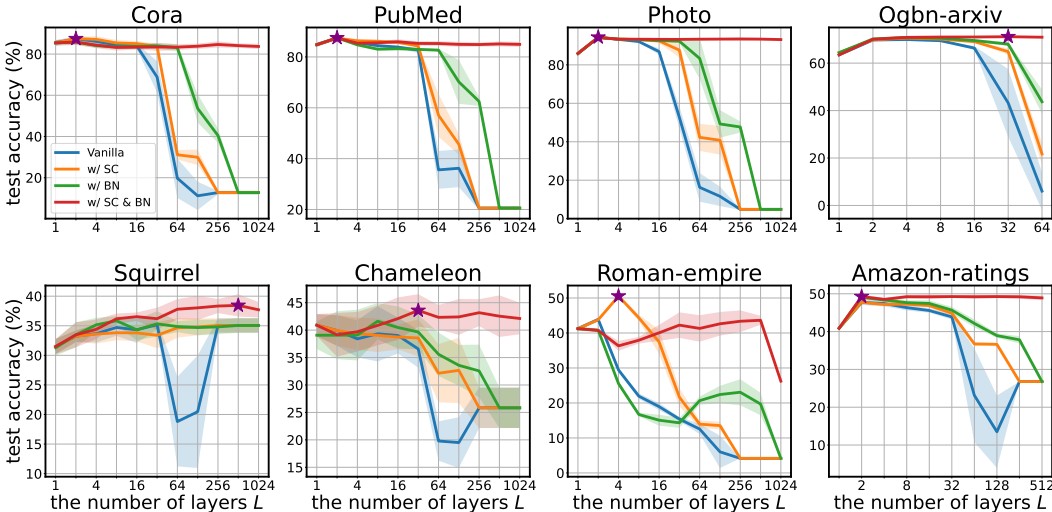

Figure 6: Effectiveness evaluation of skip connections and batch normalization on GCN. We report the test accuracy for four cases: the original model (Vanilla), the model with skip connections (w/ SC), the model with batch normalization (w/ BN), and the model with both skip connections and batch normalization (w/ SC & BN). Note that GPU memory constraints limit experiments on certain datasets to fewer layers. A star sign represents the best accuracy.

$\hat{\mathbf{A}}_{\mathrm{GAT}}$ performs weighted summation over the identical embeddings. Hence, we can observe a similar conclusion to MLPs. One would anticipate the same with GCNs, but due to the degree-dependent structure of $\hat{\mathbf{A}}_{\mathrm{GCN}}$, the behavior differs slightly from those of GATs and MLPs.

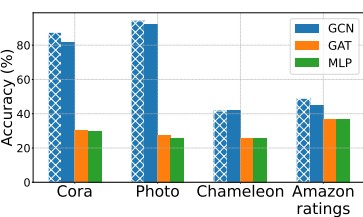

Figure 5: Test accuracy of models with all node features initialized to the same vector across nodes. The cross-hatched bar uses the original features $\mathbf{X}$, while others use uniform features $\mathbf{X}_{\mathrm{one}}$.

To make the situation more straightforward, we conduct an experiment on node classification with variants of GNNs, including GCNs, GATs, and MLPs, over four datasets. We compare the test accuracy of GNN models trained with original node features $\mathbf{X}$ versus modified features $\mathbf{X}_{\mathrm{one}}$, where all initial node features are set uniformly to match the first node feature, i.e., $(\mathbf{X}_{\mathrm{one}})_i = \mathbf{x}_1, \forall i$.

The test accuracy is summarized in Figure 5. Surprisingly, GCNs trained on $\mathbf{X}_{\mathrm{one}}$ achieve at least 91.3% of the performance trained on original features in all datasets. In the Chameleon datasets, we observe that GCNs trained on $\mathbf{X}_{\mathrm{one}}$ show even better accuracy. Figure 1 illustrates how the node embeddings change over the GCNs layer in the Cora dataset. Interestingly, all models except for the GCN fail to overcome the oversmoothing. *Based on the empirical evidence, we claim that GCNs are capable of escaping the oversmoothing regime.*

## 4 DEEPGNNS: THE ANSWER WAS ALREADY THERE, WAITING TO BE FOUND

### 4.1 TRAINING GNNS WITH A THOUSAND LAYERS

In this section, we show that performance degradation observed in deep GNNs can be resolved by simply applying residual connections and batch normalization, which are known to improve optimization in MLPs and Convolutional Neural Networks (CNNs). We also argue that the solution outperforms previous baselines designed to mitigate oversmoothing and enable the stacking of deeper GNN layers. Based on our findings, we recommend that skip connections and normalization be used as the default in deep GNNs, which have not received sufficient attention previously.

To evaluate the effectiveness of skip connections and batch normalization, we train GCN models under four different conditions: without skip connections or batch normalization, with skip connections only, with batch normalization only, and with both skip connections and batch normalization. Experiments are conducted on eight datasets, Cora (Sen et al., 2008), PubMed (Sen et al., 2008), Photo (McAuley et al., 2015), Ogbn-arxiv (Hu et al., 2020), Squirrel (Platonov et al., 2023; Rozemberczki et al., 2021), Chameleon (Platonov et al., 2023; Rozemberczki et al., 2021), Roman-empire (Platonov et al., 2023), and Amazon-ratings (Platonov et al., 2023), varying the number of layers from 1 to $1,024$. Due to computational constraints arising from the large size of Ogbn-arxiv and Amazon-ratings, experiments on these datasets are limited to a maximum depth of $64$ and $512$ layers, respectively. All experiments are conducted with learning rates of $0.001$, $0.005$, and $0.01$. We do not apply any regularization techniques such as weight decay or dropout, except for early stopping, to focus on the effects of skip connections and batch normalization.

The test accuracy of these variants is shown in Figure 6. The solid curves indicate the average over 10 repeated runs, and shaded regions indicate the standard deviation around the average. Both skip connections and batch normalization effectively alleviated performance degradation in GNNs. Furthermore, when applied together, these methods maintained stable performance even as the model depth increased. We provide the results with GAT in Appendix D.2.

We also compare the effectiveness of skip connections and batch normalization against other methods designed to mitigate oversmoothing and enable deeper GNN architectures. Specifically, we evaluate three baseline methods: DropEdge (Rong et al., 2020), PairNorm (Zhao & Akoglu, 2020), and SkipNode (Lu et al., 2024), implemented with GCNs. The results presented in Figure 7 indicate that GCNs equipped with skip connections and batch normalization maintain the most consistent test accuracy across all datasets as the number of layers increases. We provide the additional results on other datasets and details about hyperparameter selection in Appendix D.4 and Appendix E, respectively.

## 4.2 DISCUSSION

Why has the oversmoothing research community missed such an obvious result? We attempt to understand the cause by revisiting it from a historical perspective. The appendix of the celebrated GCN paper (Kipf & Welling, 2017) includes the following statement: *"For the datasets considered here,[2] best results are obtained with a 2- or 3-layer model."* This remark does not describe a general characteristic of GNN architectures but rather behavior observed in GNNs for specific datasets. However, since the subsequent introduction of oversmoothing (Keriven, 2022; Li et al., 2018) and demonstration of its exponential occurrence (Oono & Suzuki, 2020; Wu et al., 2023a) provided such a convincing explanation for the observation, later researchers appear to have prematurely concluded that GCNs inherently achieve optimal performance at limited depth. This perception led people to regard problems caused by vanishing gradients as unique characteristics of GNNs, reinforcing the misunderstanding that GNNs experience severe performance degradation with increasing depth.

We demonstrate that performance degradation is not a phenomenon specific to GNNs, and it can be resolved by addressing the vanishing gradient problem. Yet, the initially raised question of whether GNNs inherently achieve optimal performance at limited depth remains open. Even when using skip connections and batch normalization, many datasets achieve best performance at shallow depths. However, there are cases where high performance is achieved at relatively deeper depths. In Figure 6, the GCNs trained on the Squirrel, Chameleon, and Ogbn-arxiv datasets exhibit improved performance with increased depth, achieving their best performance at relatively deeper layers.

We hypothesize that the optimal depth of GNNs varies depending on dataset characteristics. Squirrel and Chameleon are heterophilic datasets known for the importance of capturing long-range dependencies (Rusch et al., 2023a). Ogbn-arxiv dataset is a large-scale dataset requiring high expressivity (Hu et al., 2020). Furthermore, previous studies (Li et al., 2019; 2021) report achieving optimal performance at deeper layers when using large-scale datasets such as the 3D point cloud and Ogbn-proteins datasets, which are not common benchmarks in the literature.

---

[2]Cora, Citeseer, and PubMed

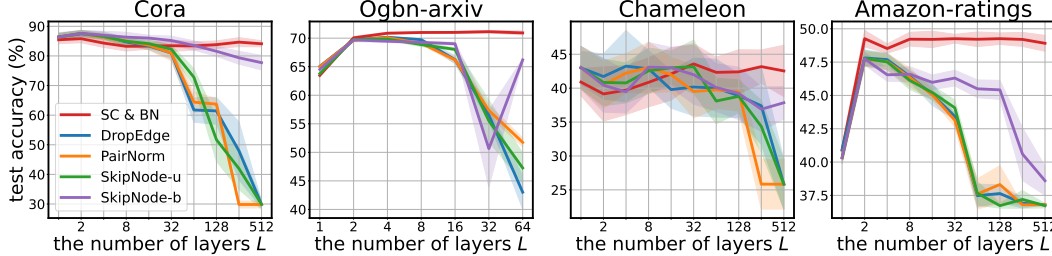

Figure 7: Test accuracy of GCN with different over-smoothing mitigation methods. We evaluate the performance of skip connection with batch normalization (SC & BN) with three baseline methods: DropEdge, PairNorm, and SkipNode. SkipNode-u and SkipNode-b are two variants of SkipNode with uniform sampling and biased sampling, respectively.

To support our hypothesis, we provide additional interesting results in Figure 8, which shows test accuracy of GCNs on Cora and PubMed, whose features are modified to $\mathbf{X}_{one}$, varying the number of layers from 1 to 1024. The results show a trend of performance improvement with increased depth.

Our interpretation is that a GCN trained on $\mathbf{X}_{one}$ likely requires high expressivity, as the model should reconstruct the information loss from the original data. We provide the result with other datasets in Appendix D.3.

Taken together, we conjecture that the widespread belief that GNNs perform optimally with only a few layers has become entrenched due to a series of overlapping misunderstandings and oversights. Specifically, we hypothesize that initialization issues and the resulting zero-collapsing, along with inherent characteristics of the most commonly used benchmark datasets such as Cora, CiteSeer, and PubMed, have inadvertently shaped the current research landscape.

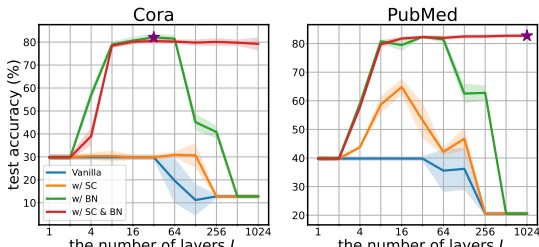

Figure 8: Test accuracy of GCNs with uniform features $\mathbf{X}_{one}$ on two datasets, Cora and PubMed. A star indicates the best accuracy.

## 5 CONCLUSION

Based on our findings, we conclude that oversmoothing is an overestimated problem, not being a main obstacle for deep GNNs. We claim that the real problem is vanishing gradient, a classical problem in neural networks. Our conclusion aligns with several observations reported in previous literature. For instance, Zhou et al. (2021b) and Zhang et al. (2022) show no significant performance degradation when repeatedly applying the aggregation process without the transformation step, suggesting that oversmoothing alone might not fundamentally cause performance decline. Experiments by Rusch et al. (2023a) demonstrate performance degradation even when oversmoothing was successfully mitigated. Furthermore, various studies (Li et al., 2018; Wu et al., 2023b; Lu et al., 2024; Arroyo et al., 2025) have indicated that the depth-related limitations of GNNs originate primarily from optimization challenges rather than oversmoothing itself. Notably, Li et al. (2019) successfully trained a 56-layer GCN by adapting residual connections. Similarly, Li et al. (2021) achieve the best performance using a 448-layer GNN, primarily by addressing memory consumption issues. However, none of these studies fully challenged the belief that oversmoothing is one of the main reasons hindering the construction of deep GNNs.

We consider that the potential benefits of deep GNNs remain largely unexplored, due to the prevailing belief that GNN performance deteriorates at deeper layers. Our examination of publicly available GNN models reveals that hyperparameter search spaces for tuning network depth are predominantly restricted to fewer than 10 layers (Bo et al., 2021; He et al., 2021; Luo et al., 2024; Pei et al., 2020; Platonov et al., 2023). Even when employing GNNs on LRGB datasets that require long-range interaction, the depth is typically limited to fewer than six layers (Dwivedi et al., 2022).

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

## A  ADDITIONAL RELATED WORK

Many studies have attempted to enable deeper stacking of layers without performance degradation by mitigating oversmoothing. PairNorm (Zhao & Akoglu, 2020) maintains total pairwise embedding distances constant through normalization. DropEdge (Rong et al., 2020) randomly removes edges from the graph, thus delaying oversmoothing. Additionally, DropMessage (Fang et al., 2023) unifies and improves random dropping in GNNs by directly masking elements in the message matrix, achieving better generalization, stability, and over-smoothing mitigation. Energetic Graph Neural Networks (Zhou et al., 2021a) incorporate Dirichlet energy into the loss function as a regularizer. Ordered GNN (Song et al., 2023) separately preserves aggregated embeddings within specific hops. Gradient Gating (Rusch et al., 2023b) introduces a mechanism to stop learning individually for each node before local oversmoothing occurs. Park et al. (2024) propose the framework that can reverse the aggregation process. GCNII (Chen et al., 2020) introduces initial residual connections and identity mapping to solve oversmoothing. LSGAT (Su et al., 2024) adaptively adjusts attention coefficients across layers to avoid oversmoothing. MTGCN (Pei et al., 2024) identifies the root cause of oversmoothing as information loss due to heterophily mixing in aggregation and mitigates it by separating message passing into multiple semantic tracks. SkipNode (Lu et al., 2024) alleviates over-smoothing and gradient vanishing in deep GCNs by selectively skipping message passing for certain nodes. Some studies propose to change the dynamics of GNNs, given that classical GNNs resemble diffusion-like dynamics (Chamberlain et al., 2021). GRAND++ (Thorpe et al., 2022) uses a source term to prevent the convergence of the embedding. PDE-GCN (Eliasof et al., 2021) and Graph-coupled oscillator networks (Rusch et al., 2022) are designed based on the wave function and nonlinear oscillator structures, respectively. However, only a few methods have succeeded in achieving performance improvements with increasing depth.

## B  NODE SIMILARITY MEASURES

In this section, we provide four node similarity measures used in previous works for completeness and how the exponential oversmoothing is characterized via the similarity measures.

Oono & Suzuki (2020) uses distance between embedding $\mathbf{X}$ and a space $\mathcal{M} = \{\tilde{\mathbf{D}}^{\frac{1}{2}}\mathbf{1}_N \otimes \mathbf{w} | \mathbf{w} \in \mathbb{R}^d\} \in \mathbb{R}^{N \times d}$ as a node similarity measure, i.e.:

$$\mu_{\texttt{oono}}(\mathbf{X}) \coloneqq \inf\{\|\mathbf{X} - \mathbf{Y}\|_F \,|\, \mathbf{Y} \in \mathcal{M}\}\,, \tag{A1}$$

where $\mathbf{1}_N = [1, \cdots, 1]^\top \in \mathbb{N}^N$, $\otimes$ denotes Kronecker product, and $\|\cdot\|_F$ indicates Frobenius norm. Embedding $\mathbf{X}$ spans on $\mathcal{M}$, i.e., $\mu_{\texttt{oono}}(\mathbf{X}) = 0$, if and only if $\mathbf{X}$ satisfies Proposition 1.

For demonstration of the exponential occurrence of oversmoothing in GCN, they show that

$$\mu_{\texttt{oono}}(\mathbf{X}^{(\ell)}) \leq (s\lambda)^\ell \mu_{\texttt{oono}}(\mathbf{X}^{(0)})\,, \tag{A2}$$

where $s$ is the largest operator norm of $\mathbf{W}^{(\ell)}, \forall \ell$, $\lambda < 1$ is the second largest eigenvalue of $\hat{\mathbf{A}}_{\text{GCN}}$. In the process of demonstration, they show how each aggregation, transformation, and activation step influences the oversmoothing as follows:

$$\mu_{\texttt{oono}}(\hat{\mathbf{A}}_{\text{GCN}}\mathbf{X}) \leq \lambda\mu_{\texttt{oono}}(\mathbf{X}),\ \mu_{\texttt{oono}}(\mathbf{X}\mathbf{W}) \leq \|\mathbf{W}\|_2\,\mu_{\texttt{oono}}(\mathbf{X}),\ \mu_{\texttt{oono}}(\sigma(\mathbf{X})) \leq \mu_{\texttt{oono}}(\mathbf{X}), \tag{A3}$$

where $\|\cdot\|_2$ denotes operator norm. Equation (A3) suggests that exponential oversmoothing occurs due to the aggregation and transformation step.

Empirical results show that $\mu_{\texttt{oono}}(\mathbf{X}^{(\ell)})$ exponentially decays in practice, satisfying the inequality provided by the theory. However, an adequate explanation of why the distance continues to decay even in cases where the upper bound explodes due to the condition $s\lambda > 1$ was not provided.

Cai & Wang (2020) simplify the proof of Oono & Suzuki (2020) by using Dirichlet energy as a measure of oversmoothing, i.e.,

$$\mu_{\texttt{cai}}(\mathbf{X}) \coloneqq \frac{1}{2}\sum_{i \in \mathcal{V}}\sum_{j \in \mathcal{N}_i}\left\|\frac{\mathbf{x}_i}{\sqrt{1+d_i}} - \frac{\mathbf{x}_j}{\sqrt{1+d_j}}\right\|_F^2. \tag{A4}$$

$\mu_{\texttt{cai}}(\cdot) = 0$ also holds if and only if [Proposition 1](). They prove that $\mu_{\texttt{cai}}(\mathbf{X}^{(\ell)}) \leq (s(1 - \bar{\lambda})^2)^{\ell} \mu_{\texttt{cai}}(\mathbf{X}^{(0)})$ by showing:

$$\mu_{\texttt{cai}}(\hat{\mathbf{A}}_{\texttt{GCN}}\mathbf{X}) \leq (1 - \bar{\lambda})^2 \mu_{\texttt{cai}}(\mathbf{X}), \ \mu_{\texttt{cai}}(\mathbf{X}\mathbf{W}) \leq \|\mathbf{W}\|_2 \, \mu_{\texttt{cai}}(\mathbf{X}), \ \mu_{\texttt{cai}}(\sigma(\mathbf{X})) \leq \mu_{\texttt{cai}}(\mathbf{X}), \tag{A5}$$

where $\bar{\lambda}$ indicates smallest non-zero eigenvalue of augmented normalized Laplacian $\mathbf{I}_N - \hat{\mathbf{A}}_{\texttt{GCN}}$. The demonstration process also implies that the aggregation and transformation step causes the exponential oversmoothing.

Wu et al. (2023a) extend the previous theoretical analyses to demonstrate that GATs also lose expressive power exponentially. They define a node similarity measure:

$$\mu_{\texttt{wu}}(\mathbf{X}) := \left\| \mathbf{X} - \frac{1}{N} \mathbf{1}_N \mathbf{1}_N^{\top} \mathbf{X} \right\|_F , \tag{A6}$$

where $\mu_{\texttt{wu}}(\mathbf{X}) = 0$ if and only if [Proposition 2]() holds, instead of [Proposition 1](). They show that $\mu_{\texttt{wu}}(\mathbf{X}^{(\ell)})$ approaches zero at an exponential rate as passing the GAT layers, explaining that the exponential oversmoothing occurred due to the joint spectral radius of a $\hat{\mathbf{A}}_{\texttt{GAT}}$ being less than one. They validate their theory by numerical experiments, showing that $\mu_{\texttt{wu}}(\mathbf{X}^{(\ell)})$ exponentially converges to zero in practice.

Rusch et al. (2023a) introduce one more node similarity measure:

$$\mu_{\texttt{rusch}}(\mathbf{X}) := \sqrt{\frac{1}{N} \sum_{i \in \mathcal{V}} \sum_{j \in \mathcal{N}_i} \|\mathbf{x}_i - \mathbf{x}_j\|_F^2} , \tag{A7}$$

which is also called Dirichlet energy but has a different form with $\mu_{\texttt{cai}}(\mathbf{X})$. $\mu_{\texttt{cai}}(\cdot)$ satisfies $\mu_{\texttt{wu}}(\mathbf{X}) = 0$ if and only if [Proposition 2]() holds. Although the measure was not utilized in theoretical analysis, Rusch et al. (2023a) provide the experimental results that $\mu_{\texttt{rusch}}(\mathbf{X}^{(\ell)})$ converges exponentially to zero when $\ell$ increases in GCN, GAT, and GraphSAGE (Hamilton et al., 2017).

## C  DATASET STATISTICS

We provide detailed statistics and explanations about the dataset used for the experiments in Table A1 and the paragraphs below.

Table A1: Statistics of the datasets utilized in the experiments.

| Dataset | # nodes | # edges | # features | # classes |
|---|---|---|---|---|
| Cora | 2,708 | 5,278 | 1,433 | 7 |
| CiteSeer | 3,327 | 4,552 | 3,703 | 6 |
| PubMed | 19,717 | 44,324 | 500 | 3 |
| Computers | 13,752 | 245,861 | 767 | 10 |
| Photo | 7,650 | 119,081 | 745 | 8 |
| Ogbn-arxiv | 169,343 | 1,166,243 | 128 | 40 |
| Squirrel | 2,223 | 46,998 | 2,089 | 5 |
| Chameleon | 890 | 8,854 | 2,325 | 5 |
| Roman-empire | 22,662 | 32,927 | 300 | 18 |
| Amazon-ratings | 24,492 | 93,050 | 300 | 5 |

**Cora, CiteSeer, and PubMed**  Each node represents a paper, and an edge indicates a reference relationship between two papers. The task is to predict the research subjects of the papers.

**Computers and Photo**  Each node represents a product, and an edge indicates a high frequency of concurrent purchases of the two products. The task is to predict the product category.

**Ogbn-arxiv**   Each node represents a Computer Science arXiv paper, and each directed edge indicates a citation between papers. The task is to predict the subject area of each paper.

**Squirrel and Chameleon**   Each node represents a Wikipedia page, and an edge indicates a link between two pages. The task is to predict the monthly traffic for each page. We use the classification version of the dataset, where labels are converted by dividing monthly traffic into five bins. We adopted the filtering process to prevent train-test data leakage as recommended by (Platonov et al., 2023).

**Roman-empire**   Each node represents a word extracted from the English Wikipedia article on the Roman Empire, and an edge indicates a grammatical or sequential relationship between words. The task is to predict the part-of-speech tag of each word.

**Amazon-ratings**   Each node represents a product from the Amazon co-purchasing network, and an edge indicates a frequent co-purchase between two products. The task is to predict the product category based on user co-purchasing patterns.

## D   SUPPLEMENTARY EXPERIMENTAL RESULTS

In this section, we provide the supplementary experimental results for the main paper.

### D.1   SUPPLEMENTARY EXPERIMENTAL RESULTS FOR THE CHANGE OF NODE SIMILARITY MEASURE

We provide the supplementary experimental results for the change of node similarity measure. The results of the eight datasets, CiteSeer, PubMed, Computers, Photo, Ogbn-arxiv, Squirrel, Roman-empire, and Amazon-ratings, on GCNs and GATs are provided in Figure A1 and Figure A2, respectively.

### D.2   SUPPLEMENTARY EXPERIMENTAL RESULTS FOR THE EFFECTIVENESS EVALUATION OF THE SKIP CONNECTION AND BATCH NORMALIZATION WITH GATS

We provide supplementary results for the effectiveness evaluation of the skip connection and batch normalization with GATS in Figure A3, which illustrates test accuracy on eight datasets: CiteSeer, PubMed, Computers, Photo, Ogbn-arxiv, Squirrel, Roman-empire, and Amazon-ratings.

### D.3   SUPPLEMENTARY EXPERIMENTAL RESULTS ON MODIFIED FEATURE $\mathbf{X}_{\text{one}}$

Supplementary experimental results on modified feature $\mathbf{X}_{\text{one}}$ on additional eight datasets, CiteSeer, Computers, Photo, Ogbn-arxiv, Squirrel, Chameleon, Roman-empire, and Amazon-ratings, are provided in Figure A4.

### D.4   SUPPLEMENTARY EXPERIMENTAL RESULTS OF TEST ACCURACY COMPARISON BETWEEN BASELINES

Supplementary experimental results of test accuracy comparison among the baselines designed to mitigate oversmoothing on an additional six datasets, CiteSeer, PubMed, Photo, Computers, Squirrel, and Roman-empire, are provided in Figure A5.

### D.5   SUPPLEMENTARY EXPERIMENTAL RESULTS WITH HE INITIALIZATION

We demonstrate that He initialization can mitigate exponential oversmoothing in GNNs. Figure A6 and Figure A7 show the changes of the node similarity measure across layers in GCN and GAT, respectively, with He initialization adopted. We observe that the decay rate significantly decreases compared to results with Glorot initialization shown in Figure 3. The distribution of node embedding values across four selected layers of GCN and GAT is visualized in Figure A8. The results confirm that He initialization mitigates zero-collapsing compared to Glorot initialization shown in Figure 4.

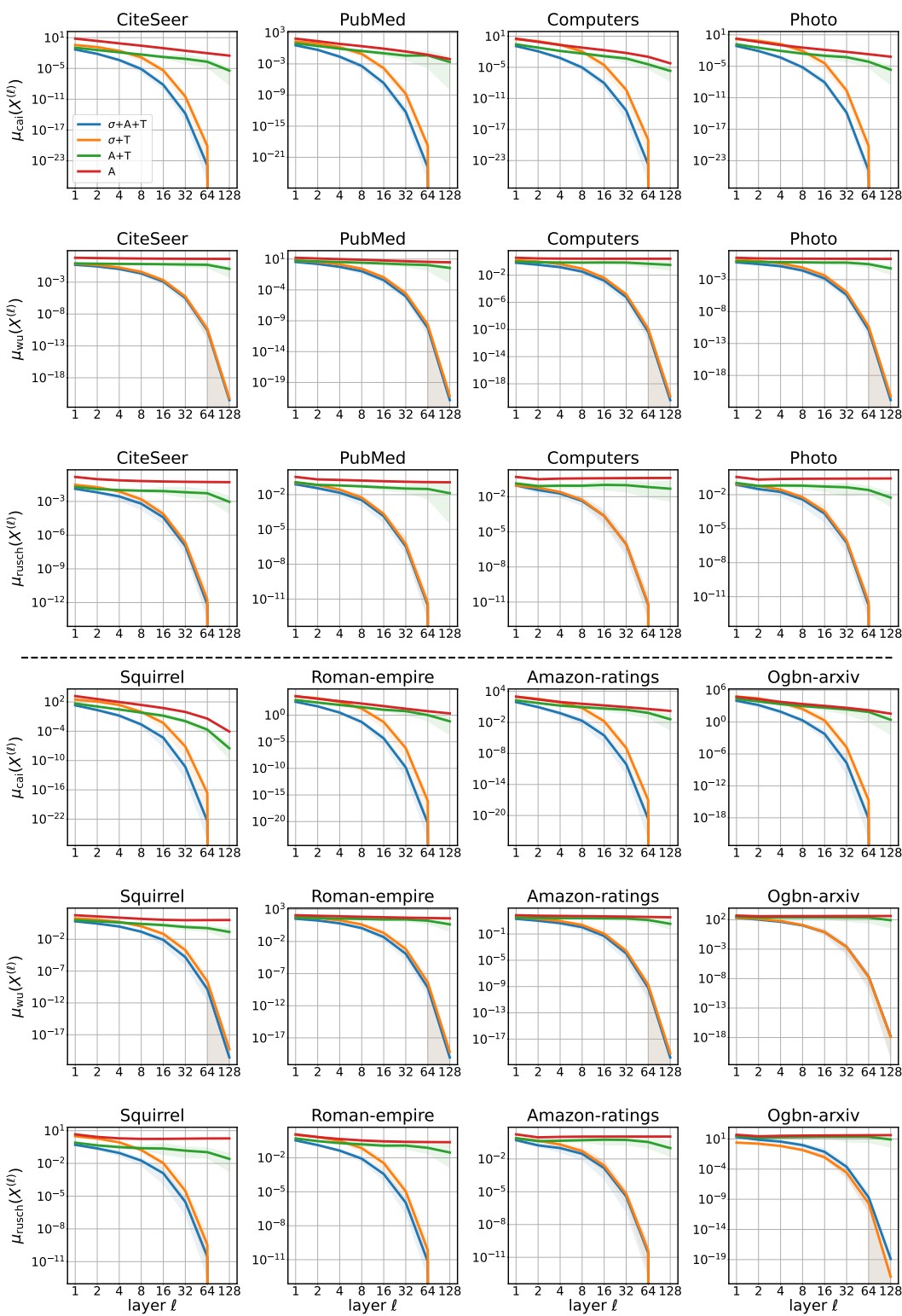

Figure A1: Oversmoothing levels measured by three metrics across a 128-layer GCN. For each model, we compare four variants based on combinations of aggregation (A), transformation (T), and activation ($\sigma$) steps.

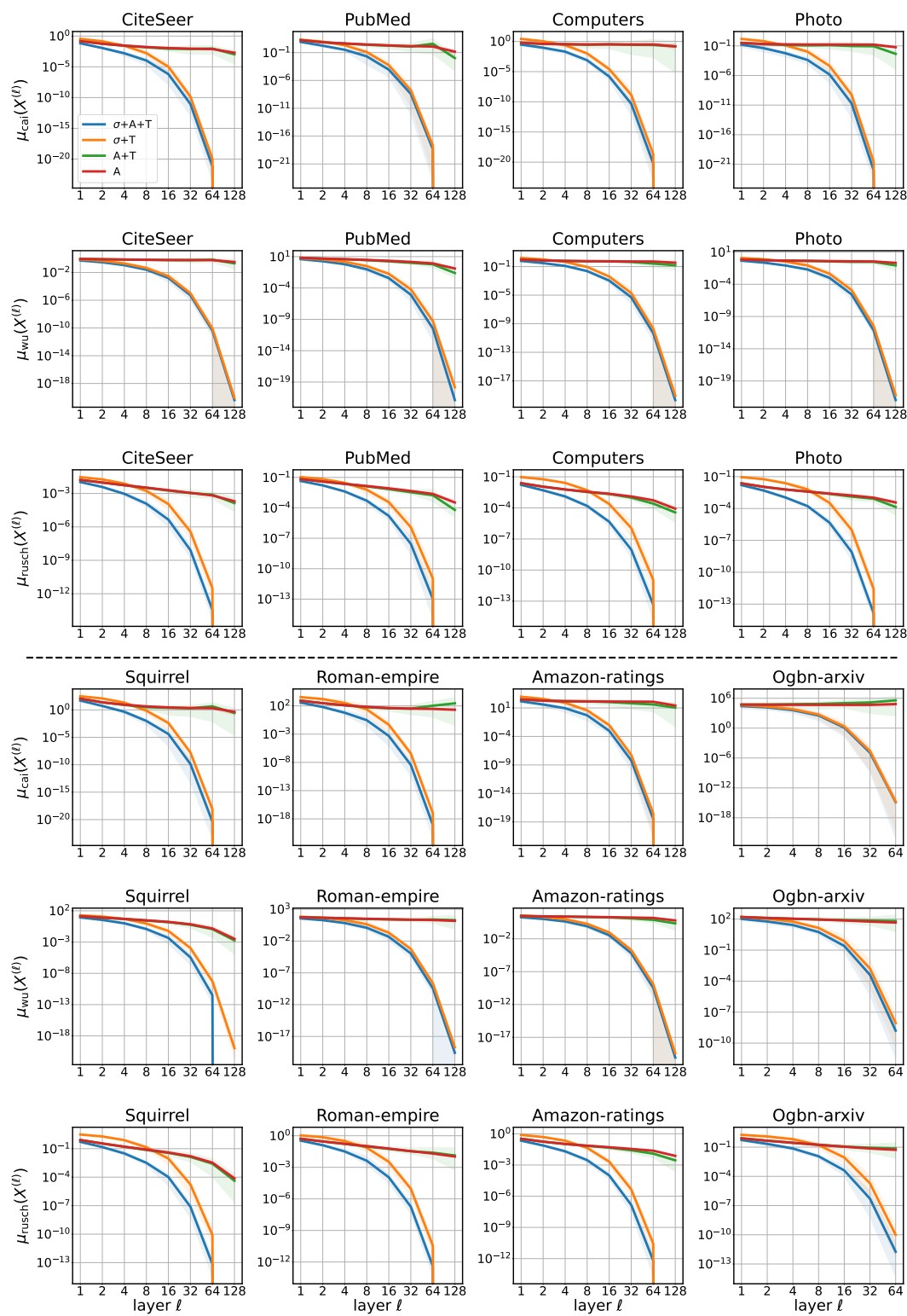

Figure A2: Oversmoothing levels measured by three metrics across a 128-layer GAT. For each model, we compare four variants based on combinations of aggregation (A), transformation (T), and activation ($\sigma$) steps.

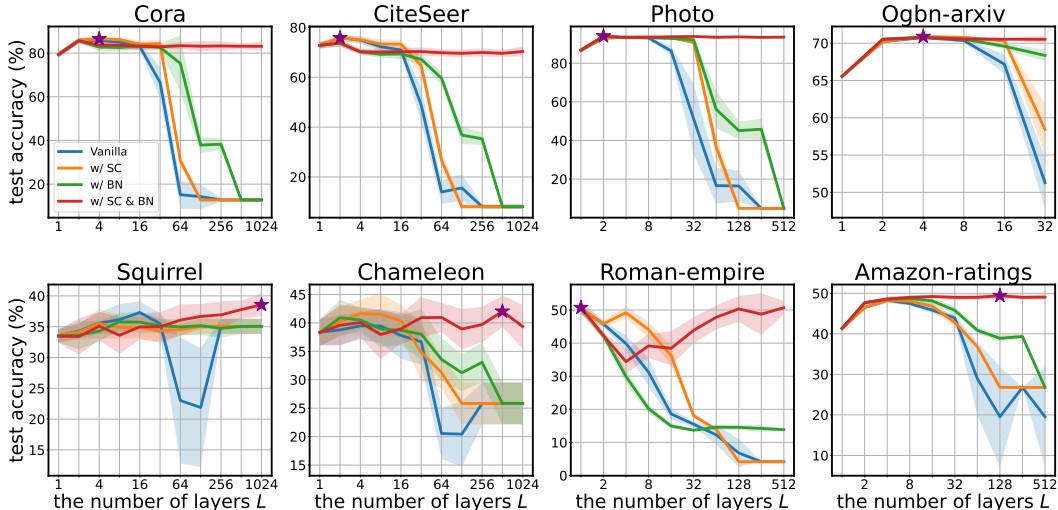

Figure A3: Effectiveness evaluation of skip connections and batch normalization on GAT. We report the test accuracy for four cases: the original model (Vanilla), the model with skip connections (w/ SC), the model with batch normalization (w/ BN), and the model with both skip connections and batch normalization (w/ SC & BN).

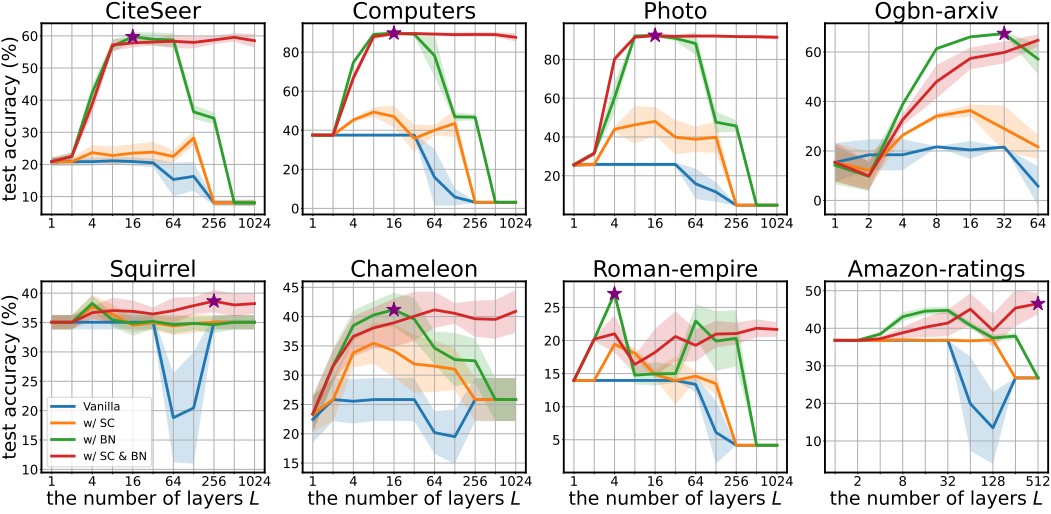

Figure A4: Test accuracy of GCNs with uniform features $\mathbf{X}_{\text{one}}$ on the remaining eight datasets.

# E  EXPERIMENTAL DETAILS

In this section, we describe the details of our training setup for experiments in Section 3.2, Section 3.3, and Section 4.1. Our experiments were conducted on an AMD EPYC 7513 32-core processor and a single NVIDIA RTX A6000 GPU with 48GB of memory.

To obtain the results presented in Figure 3 in Section 3.2, we measure node similarity without training, since a 128-layer GNN cannot be trained due to the vanishing gradient issue described in the main paper. We utilize the 10 existing standard train/validation/test splits for all datasets. Following Wu et al. (2023a), we set the hidden dimension to 32. The same experimental setup is adopted for Figure A1 and Figure A2, except for the GAT experiment on the Ogbn-arxiv dataset. Due to memory limitations, we set the hidden dimension to 16 in this case.

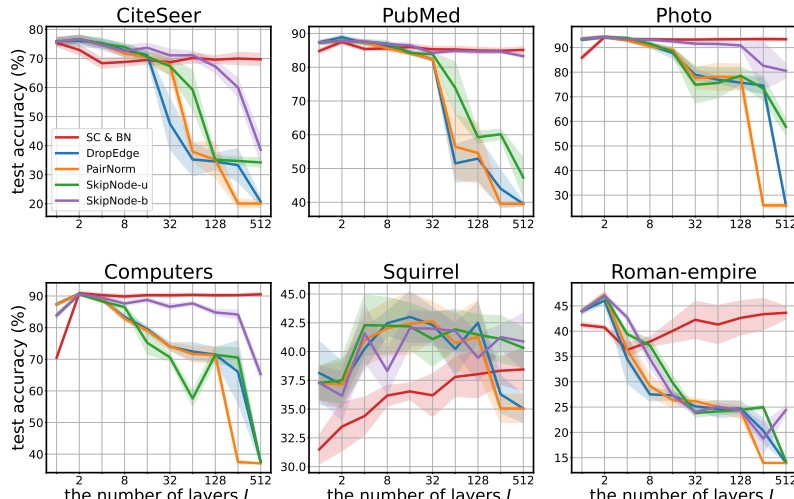

Figure A5: Test accuracy of GCN with different over-smoothing mitigation methods. We evaluate the performance of skip connection with batch normalization (SC & BN) with three baseline methods: DropEdge, PairNorm, and SkipNode. SkipNode-u and SkipNode-b are two variants of SkipNode with uniform sampling and biased sampling, respectively.

To obtain results shown in Figure 5 in Section 3.3, we use the 10 existing standard train/validation/test splits for all datasets. We train for a maximum of $1,000$ epochs using early stopping on the validation set with a patience of 200 epochs. For each model, we perform a learning rate search within $\{0.001, 0.005, 0.01\}$ and depth search within $\{1, 2, 4, \cdots, 1,024\}$ and select whether to use skip connections and batch normalization based on validation performance. We fix the hidden dimension to $64$.

The above settings are also adopted for the experiments described in Section 4.1. For baseline methods, we tune the learning rate within the range $\{0.001, 0.005, 0.01, 0.05, 0.1\}$. For other hyperparameters, PairNorm's settings follow the original recommendations. For DropEdge and PairNorm, we vary the sampling rates within $\{0.05, 0.1, ..., 0.9\}$.

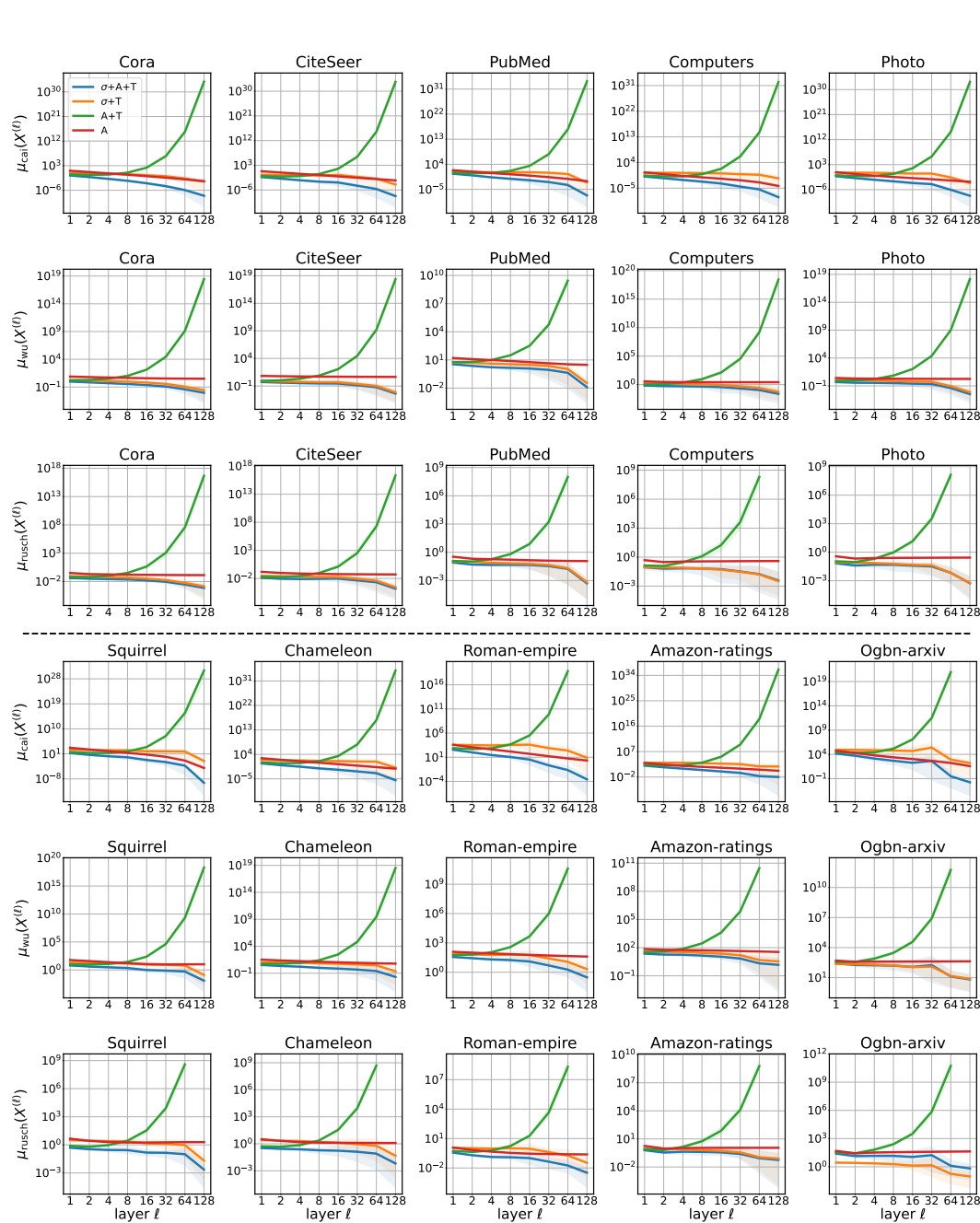

Figure A6: Oversmoothing levels measured by three metrics across 128-layer GCN model evaluated on all datasets with He initialization.

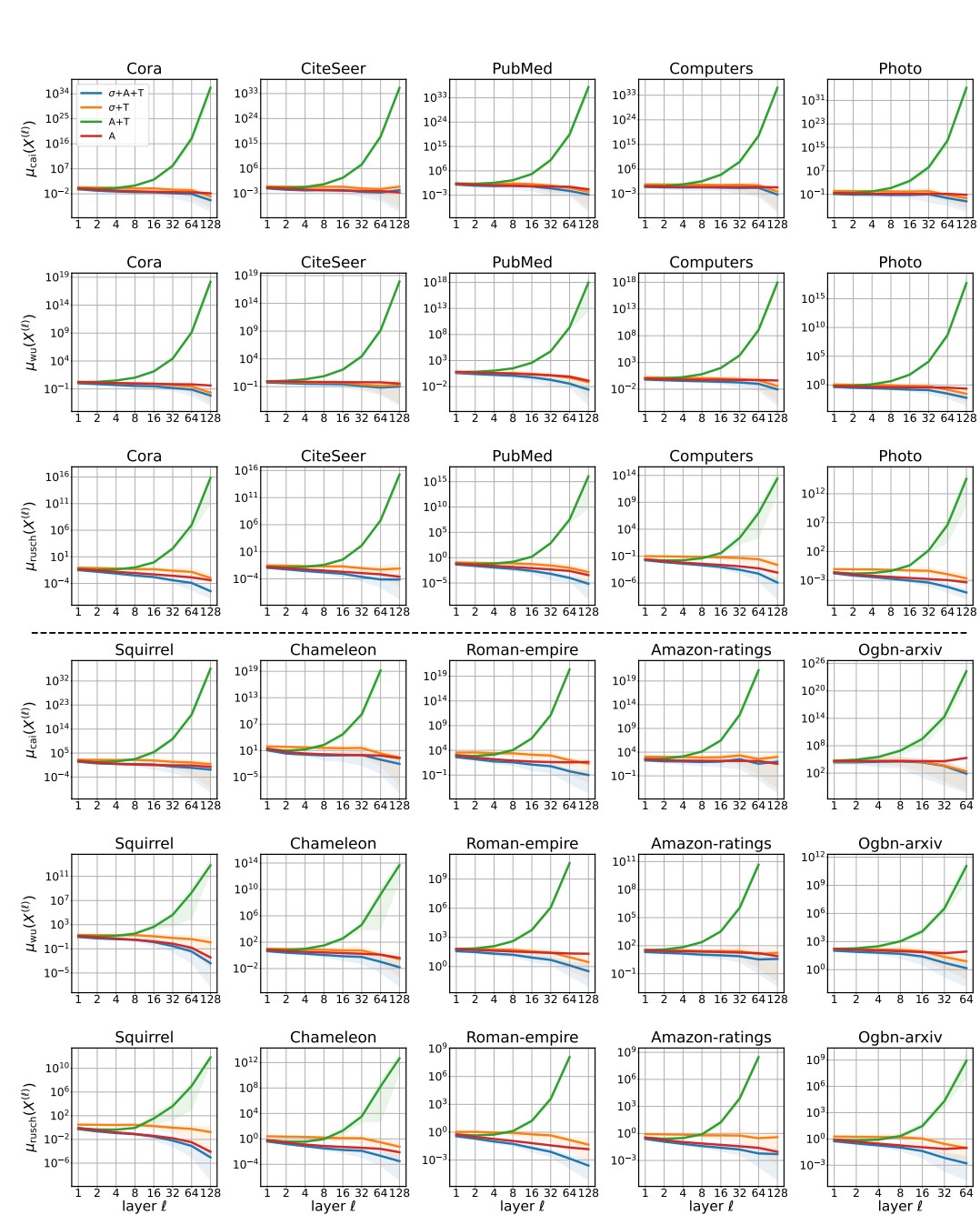

Figure A7: Oversmoothing levels measured by three metrics across 128-layer GAT model evaluated on all datasets with He initialization.

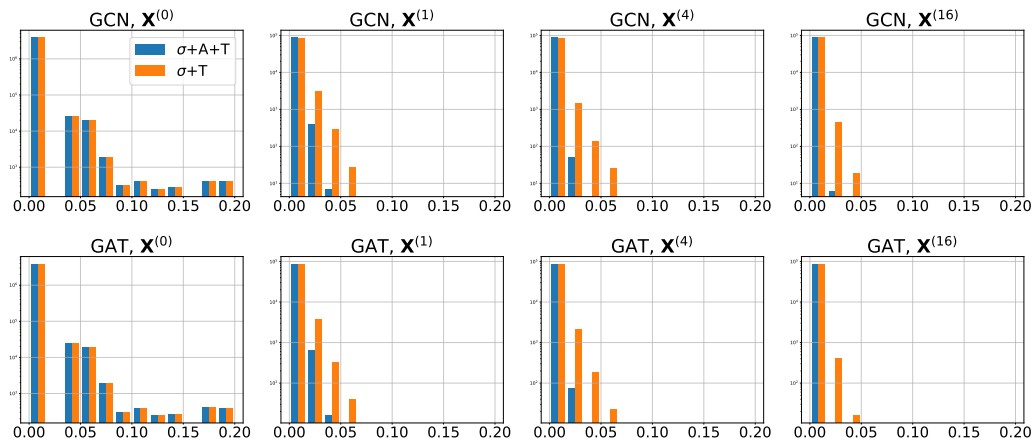

Figure A8: Histogram of node embedding values $\mathbf{X}^{(\ell)}$ from four selected layers, with weights initialized using He initialization.

