# OpenReview forum: "The Oversmoothing Fallacy: A Misguided Narrative in GNN Research"
_ICLR.cc/2026/Conference — Submitted to ICLR 2026_

### Official Review · Reviewer_Njko · 2025-10-21

**Soundness:** 1
**Presentation:** 3
**Contribution:** 1
**Rating:** 2
**Confidence:** 4

**Summary:**

The paper argues that the performance degradation in deep GNNs has been mistakenly attributed to oversmoothing, while the real problem in practice is “zero-collapsing” and vanishing gradients arising from the transformation+activation steps. This paper  (i) separates aggregation, linear transformation, and activation, (ii) measures oversmoothing with several metrics across different scenarios (σ+A+T, σ+T, A+T, A), and (iii) claims aggregation has only a marginal role. This work further shows that residual/skip connections and batch normalization enable very deep GNNs without degradation, and position asymptotic oversmoothing results as complementary but not explanatory for finite-depth models.

**Strengths:**

- The paper is well written.
- This paper is a good reminder that optimization is also important in GNNs (vanishing gradients).

**Weaknesses:**

- Even though this paper is a good reminder that optimization is important, it doesn’t bring anything new about the oversmoothing problem in the literature. For example, the following statement is already known and well studied theoretically: “Furthermore, we demonstrate that classical solutions such as skip connections and normalization enable the successful stacking of deep GNN layers without performance degradation”. See [R2]. From a practical perspective, the following claim has been observed empirically since 2019: “Remarkably, we show that by properly integrating these two simple yet effective strategies, it is feasible to train extremely deep GNNs, successfully scaling up to 1,024 layers”. See [R3]. Solving oversmoothing by using residual connections does not mean that oversmoothing is caused by vanishing gradients. This is a logical fallacy; correlation does not imply causation.
- The following claim is not true: “Our finding challenges prior beliefs about oversmoothing being unique to GNNs”. Previous works have reported oversmoothing in other architectures, e.g., transformers [R1].
- Regarding the remark in the introduction, I do not get the point. This remark says this work is not in contradiction with asymptotic analyses and that practical issues are dominated by vanishing gradients; yet, elsewhere, the narrative suggests the community has “misunderstood” oversmoothing and overestimated the aggregation’s role. The take-home message from this paper oscillates between “oversmoothing exists asymptotically but is not the practical bottleneck” and “aggregation’s role is marginal and oversmoothing has been overestimated” (too broad without stronger causal evidence).
- In Figure 3, this paper argues that “oversmoothing is observable without the aggregation step”. I disagree with this statement; again, correlation does not imply causation.
- Typo, I guess: “Note that A+T is equal to the MLP architecture”.
- I don’t understand Figure 4. Is this just a histogram of all values in the embeddings? Oversmoothing should be analyzed column-wise in $X^{(l)}$. The argument in the literature is that each of these columns (also known as graph signals) converges to a stationary state, also characterized by a zero Dirichlet energy. Plotting such a histogram does not make sense.
- While this paper attributes the oversmoothing problem primarily to vanishing gradients and argues that the aggregation operator plays only a marginal role, Roth and Liebig (2023) [R4] theoretically demonstrate that the spectral properties of the aggregation operator are, in fact, the main cause, driving node representations into a low-dimensional subspace irrespective of the feature transformations. From that perspective, the residual connections proposed here do not resolve vanishing gradients per se, but rather correspond to a specific implementation of a Sum of Kronecker Products to alleviate rank collapse. Besides, the following statement in that remark seems to be false: “These results provide valuable insight into the limiting behavior of message-passing networks, but they largely characterize asymptotic convergence rather than explaining the degradation observed at practical depths”. Indeed, [R4] provided empirical evidence for “practical depths” (see figures 2 and 4 in [R4]).

**General comment**: The main claim of this paper goes against much of the existing theoretical and experimental research, but the authors do not provide a new theory to support this different view. The experiments shown are not strong enough to question or replace the current understanding of oversmoothing, and some results do not fit the authors’ own explanations. At the very least, a clear and convincing theoretical framework would be needed before such strong claims can be accepted.

---
[R1] “Mitigating Over-smoothing in Transformers via Regularized Nonlocal Functionals”, NeurIPS 2023.
[R2] “Residual Connections and Normalization Can Provably Prevent Oversmoothing in GNNs”, ICLR 2025.
[R3] “DeepGCNs: Can GCNs Go as Deep as CNNs?”, ICCV 2019.
[R4] “Rank Collapse Causes Over-Smoothing and Over-Correlation in Graph Neural Networks”, LoG 2023.

**Questions:**

In Figure 3, results for aggregation only (A) contradict all results about oversmoothing in the literature. This is basically an SGC model where the power of the adjacency matrix (for GCN) is $l$. We have enough theoretical evidence to say that the node embeddings should converge to a stationary distribution when the graph is connected and not bipartite. It seems to me this is a one-layer SGC with multiple linear layers. How is this experiment performed? Could you compute tr(X’LX) (Dirichlet energy)? It should converge to 0 quite fast.

---

> ### Author Response · Authors · 2025-12-03
>
> We thank the Reviewer for their careful and thoughtful review, and for recognizing (1) the clarity of our presentation and (2) that our work serves as a helpful reminder of the importance of optimization in deep GNNs. We address the Reviewer’s comments and questions in detail below.
>
> >W1. Even though this paper is a good reminder that optimization is important, it doesn’t bring anything new about the oversmoothing problem in the literature. For example, the following statement is already known and well studied theoretically: “Furthermore, we demonstrate that classical solutions such as skip connections and normalization enable the successful stacking of deep GNN layers without performance degradation”. See [R2]. From a practical perspective, the following claim has been observed empirically since 2019: “Remarkably, we show that by properly integrating these two simple yet effective strategies, it is feasible to train extremely deep GNNs, successfully scaling up to 1,024 layers”. See [R3]. Solving oversmoothing by using residual connections does not mean that oversmoothing is caused by vanishing gradients. This is a logical fallacy; correlation does not imply causation.
>
> We respectfully disagree that our paper does not bring anything new about the oversmoothing problem.
>
> First, while [R2] successfully demonstrates that deep GNNs can be trained by addressing vanishing gradients, it does not discuss why oversmoothing is not the primary obstacle to depth, nor does it analyze why optimization issues dominate aggregation-induced smoothing in practice. As a result, despite the availability of deep GNNs since 2019, the belief that “GNNs cannot be deep due to oversmoothing” has remained widespread, and numerous oversmoothing-mitigation methods have continued to be proposed, many of which we show in Figure 7 can be outperformed simply by applying skip connections and batch normalization.
>
> Second, our perspective differs from [R3]. That work argues that skip connections and batch normalization help alleviate oversmoothing. In contrast, we show that the primary benefit of these techniques comes from resolving vanishing gradients, not from mitigating aggregation-induced smoothing. Indeed, oversmoothing can be substantially alleviated even in vanilla GNNs simply by replacing Glorot initialization with He initialization (Section D.4). Nevertheless, even in this setting, performance degradation reappears due to vanishing gradients.
>
> Third, we would like to clarify that we do not claim “solving oversmoothing by residual connections proves that oversmoothing is caused by vanishing gradients.” Instead, our argument is that (i) the phenomenon typically interpreted as aggregation-driven oversmoothing is in fact dominated by zero-collapsing from transformation and activation, and (ii) existing theory implies that oversmoothing does not occur when the transformation step can counteract aggregation, i.e., when the weight parameters are properly updated in a healthy-gradient regime. Therefore, resolving vanishing gradients naturally removes the conditions under which oversmoothing becomes problematic, enabling the construction of deep GNNs.
>
> Finally, the novelty of our work lies in:
> * (a) contextualizing prior works that have already questioned oversmoothing as the fundamental barrier to depth (Section 5);
> * (b) identifying limitations in how existing oversmoothing theories have been empirically validated (Section 3);
> * (c) demonstrating that classical techniques alone can eliminate performance degradation and even outperform oversmoothing-specific methods (Section 4.1); and
> * (d) proposing a new research direction regarding when and why depth improves GNN performance depending on dataset characteristics (Section 4.2).
>
> We note that [R2] is indeed cited in support of point (a).
>
> >W2. The following claim is not true: “Our finding challenges prior beliefs about oversmoothing being unique to GNNs”. Previous works have reported oversmoothing in other architectures, e.g., transformers [R1].
>
> Thank you for pointing out this potentially misleading phrasing. To better reflect our intended meaning, we have revised the statement in the updated manuscript to:
> “Our findings challenge prior beliefs that oversmoothing in GNNs is dominated by the GNN-specific structure of aggregation.”

---

> > ### Author Response · Authors · 2025-12-03
> >
> > >W3. Regarding the remark in the introduction, I do not get the point. This remark says this work is not in contradiction with asymptotic analyses and that practical issues are dominated by vanishing gradients; yet, elsewhere, the narrative suggests the community has “misunderstood” oversmoothing and overestimated the aggregation’s role. / W8. **General comment:** The main claim of this paper goes against much of the existing theoretical and experimental research, but the authors do not provide a new theory to support this different view. The experiments shown are not strong enough to question or replace the current understanding of oversmoothing, and some results do not fit the authors’ own explanations. At the very least, a clear and convincing theoretical framework would be needed before such strong claims can be accepted. / Q1. In Figure 3, results for aggregation only (A) contradict all results about oversmoothing in the literature. This is basically an SGC model where the power of the adjacency matrix (for GCN) is $\ell$. We have enough theoretical evidence to say that the node embeddings should converge to a stationary distribution when the graph is connected and not bipartite. It seems to me this is a one-layer SGC with multiple linear layers. How is this experiment performed? Could you compute tr(X’LX) (Dirichlet energy)? It should converge to 0 quite fast.
> >
> > First, we would like to emphasize that **empirical results in our paper align with existing theories, and therefore do not require additional theoretical proofs.**
> >
> > To clarify this alignment, let us provide more specific details on how our results correspond to previous experiments. The existing theories in [R5] and [R6], for example,  present the upper bounds on the oversmoothing measure after applying aggregation, transformation, or activation, in terms of the oversmoothing measure before each operation (See equation A3 and A5). [R7] focuses on analyzing how the oversmoothing measure $\mu_{\text{wu}}$ evolves under the combination of aggregation and activation. However, both [R5] and [R7] have verified their claims using GNNs with all components ($\sigma$ + A + T), which correspond to the blue curves in Figure 3.
> >
> > In contrast, we decompose the GNN layer into its aggregation, transformation, and activation components to isolate the individual influence of each step. The effect of applying only aggregation, which corresponds to the multi-layer SGC rather than a one-layer SGC with multiple linear layers, can be observed in the red curves in Figure 3. These red curves show that aggregation alone induces a decay of specific measures, $\mu_{\text{cai}}$ for GCN and $\mu_{\text{rusch}}$ for GAT, although the decay occurs at a much slower rate compared to the blue curves. We note that $\mu_{\text{cai}}$ corresponds to the Dirichlet energy that the reviewer refers to.
> >
> > These findings are fully consistent with existing theory: they do not contradict prior analyses and remain compatible with them. Instead, our results reveal that earlier empirical validations have tended to underestimate the influence of transformation and activation, and to overestimate the influence of aggregation.
> >
> > > W3-2. The take-home message from this paper oscillates between “oversmoothing exists asymptotically but is not the practical bottleneck” and “aggregation’s role is marginal and oversmoothing has been overestimated” (too broad without stronger causal evidence).
> >
> > We would like to clarify that the two statements the Reviewer mentions—(i) “oversmoothing exists asymptotically but is not the practical bottleneck,” and (ii) “aggregation’s role is marginal and oversmoothing has been overestimated”—are fully compatible.
> >
> > Even when zero-collapsing is eliminated through He initialization (Section D.4), the oversmoothing measure still decays with depth, albeit much more slowly. As theoretical work suggests, if we were to construct a infinitely deep GNN with He initialization, the oversmoothing measure would eventually converge toward zero. In this extreme asymptotic regime, node features collapse into a one-dimensional subspace, gradients become uninformative, and meaningful learning is no longer possible. This is the asymptotic oversmoothing phenomenon predicted by existing theory.
> >
> > However, in the practical depth ranges typically considered, tens to a few hundred layers, we observe that the oversmoothing measure under He initialization remains far from zero. In this regime, the effect of aggregation-induced smoothing is negligible compared to optimization issues such as vanishing gradients.
> >
> > Thus, our take-home message is consistent: oversmoothing exists as an asymptotic property of graph propagation, but it is not the dominant factor limiting practical depth; prior empirical evidence has overstated the role of aggregation because it conflated aggregation with zero-collapsing.

---

> > > ### Author Response · Authors · 2025-12-03
> > >
> > > > W4. In Figure 3, this paper argues that “oversmoothing is observable without the aggregation step”. I disagree with this statement; again, correlation does not imply causation. / W6. I don’t understand Figure 4. Is this just a histogram of all values in the embeddings? Oversmoothing should be analyzed column-wise in $\mathbf{X}^{(\ell)}$. The argument in the literature is that each of these columns (also known as graph signals) converges to a stationary state, also characterized by a zero Dirichlet energy. Plotting such a histogram does not make sense.
> > >
> > > We would like to clarify why we state that “oversmoothing is observable without the aggregation step” in relation to Figures 3 and 4.
> > >
> > > We begin by recalling the definition of oversmoothing. Conceptually, oversmoothing refers to the phenomenon in which node representations become indistinguishable as layers increase. An attempt to formalize this concept mathematically was first made in [R8], where oversmoothing is defined as the exponential decay of an oversmoothing measure across layers. Under this definition, if an oversmoothing measure decreases exponentially with respect to depth, then oversmoothing is said to occur. With this definition in mind, Figure 3 shows that a vanilla MLP with ReLU activation and Glorot initialization (the orange curve) indeed exhibits exponential decay in two commonly used oversmoothing measures (e.g., $\mu_{\text{cai}}$, $\mu_{\text{rusch}}$).
> > >
> > > Figure 4 provides an explanation for why this happens. As shown there, in MLPs the feature elements converge to zero due to the combination of ReLU activation and weight values not being sufficiently far from zero, a phenomenon we refer to as zero-collapsing. This zero-collapsing causes the outputs to collapse toward zero, which in turn makes all oversmoothing measures decay exponentially, even though no aggregation is present. We note that, since every element in the feature vector collapses to zero, a column-wise analysis would also conclude that the representations are indistinguishable.
> > >
> > > Thus, given the current definitions and metrics used in the literature, oversmoothing measures cannot disentangle aggregation-induced smoothing from zero-collapsing. In this sense, and under the definitions adopted in prior work, “oversmoothing is observable without the aggregation step” simply reflects the fact that oversmoothing measures detect zero-collapsing as oversmoothing according to their mathematical formulation.
> > >
> > > >W5. Typo, I guess: “Note that A+T is equal to the MLP architecture”.
> > >
> > > Thank you for pointing out this typo. We have corrected it to $\sigma+T$ in the revised manuscript.
> > >
> > > >W7. While this paper attributes the oversmoothing problem primarily to vanishing gradients and argues that the aggregation operator plays only a marginal role, Roth and Liebig (2023) [R4] theoretically demonstrate that the spectral properties of the aggregation operator are, in fact, the main cause, driving node representations into a low-dimensional subspace irrespective of the feature transformations. From that perspective, the residual connections proposed here do not resolve vanishing gradients per se, but rather correspond to a specific implementation of a Sum of Kronecker Products to alleviate rank collapse. Besides, the following statement in that remark seems to be false: “These results provide valuable insight into the limiting behavior of message-passing networks, but they largely characterize asymptotic convergence rather than explaining the degradation observed at practical depths”. Indeed, [R4] provided empirical evidence for “practical depths” (see figures 2 and 4 in [R4]).
> > >
> > > We would like to clarify that our paper does not claim that the oversmoothing phenomenon is primarily caused by vanishing gradients. Our argument is that the exponential decay observed in prior empirical studies, often interpreted as aggregation-driven oversmoothing, is in fact dominated by zero-collapsing, which arises from the combination of initialization and activation. When zero-collapsing is removed, for example via He initialization (Section D.4), the oversmoothing measure remains far from zero even at 128-layerd depths, yet performance degradation still appears. This indicates that the observed degradation in deep GNNs is driven by vanishing gradients, not by aggregation-induced oversmoothing.
> > >
> > > The goal of our paper is therefore to point out that oversmoothing has been overestimated as the main practical limitation, and that the role of vanishing gradients has been underappreciated in the literature. Questions related to rank collapse, such as those analyzed in [R4], lie outside the scope of our work. If skip connections or batch normalization incidentally mitigate rank collapse in addition to vanishing gradients, that would indeed be an interesting direction for future research.

---

> > > > ### Author Response · Authors · 2025-12-03
> > > >
> > > > [R1] “Mitigating Over-smoothing in Transformers via Regularized Nonlocal Functionals”, NeurIPS 2023.
> > > >
> > > > [R2] “Residual Connections and Normalization Can Provably Prevent Oversmoothing in GNNs”, ICLR 2025.
> > > >
> > > > [R3] “DeepGCNs: Can GCNs Go as Deep as CNNs?”, ICCV 2019.
> > > >
> > > > [R4] “Rank Collapse Causes Over-Smoothing and Over-Correlation in Graph Neural Networks”, LoG 2023.
> > > >
> > > > [R5] Kenta Oono and Taiji Suzuki. Graph neural networks exponentially lose expressive power for node classification. In International Conference on Learning Representations, 2020.
> > > >
> > > > [R6] Chen Cai and Yusu Wang. A note on over-smoothing for graph neural networks. In ICML Graph Representation Learning and Beyond (GRL+) Workshop, 2020.
> > > >
> > > > [R7] Wu, Xinyi, et al. "Demystifying oversmoothing in attention-based graph neural networks." Advances in Neural Information Processing Systems 36 (2023): 35084-35106.
> > > >
> > > > [R8] Rusch, T. Konstantin, Michael M. Bronstein, and Siddhartha Mishra. "A survey on oversmoothing in graph neural networks." arXiv preprint arXiv:2303.10993 (2023).

---

### Official Review · Reviewer_W4ET · 2025-10-30

**Soundness:** 1
**Presentation:** 2
**Contribution:** 1
**Rating:** 2
**Confidence:** 5

**Summary:**

The paper investigates the impact of oversmoothing on the degradation of performance of graph neural networks.
The paper empirically investigates which components of the update rule contribute most to the problem in an ablation study.
The authors come to the conclusion that oversmoothing is not the main issue, but that a common problem in conventional neural networks called "dying ReLU" - a form of vanishing gradients - is the main obstacle to overcome in order to train deep GNNs.
The authors therefore propose to use conventional deep learning methods called residual connections and batch normalization to combat the problem of performance degradation and show empirically that performance degradation can be mitigated using these techniques.

**Strengths:**

- The paper asks an interesting and profound question about the strength of the impact of certain phenomena in the performance of GNNs.
- The paper displays a thorough review of related literature and identifies problems in previous manuscripts.

**Weaknesses:**

-  The main hypothesis of the paper that "Oversmoothing is not a problem for the performance of deep GNNs" is not corroborated by the evidence layed out. Theoretical considerations are missing entirely.
    - The premise that researchers believe that oversmoothing is the single most impactful problem for deep GNNs is flawed. It makes little sense to believe that the problems plaguing MLPs would not carry over to GNNs (that use MLPs internally). However, oversmoothing is a *new* phenomenon and problem that is unique to GNNs, which may be the reason it has received the attention that is has. Additionally, oversmoothing has a much more devastating effect than e.g. vanishing gradients: For any sensible weights, the model will exponentially converge to an uniformative representation of the nodes and not be able to recover the initial signal.
    - Figure 1 and Figure 5 depict the "recovery from oversmoothing" for GCN. However, there is no recovery: Initializing a GCN with all-ones does not start it off in a state of oversmoothing as it would for row-stochastic graph operators like the one used in GAT. Instead oversmoothing manifests in GCN as the exponential decay toward $\sqrt{d}$. So using $\sqrt{d}$ as the columns of $X$ would actually start GCN in a state of oversmoothing. The GCN hence does not "recover" from oversmoothing in Figure 1 - it was never in a state of oversmoothing to begin with. The same idea leads to the phenomenon depicted in Figure 5. The claim that "Based on the empirical evidence, [...] GCNs are capable of escaping the oversmoothing regime." is not a consequence of the empirical evaluation shown.
    - The authors conclude from the experiments shown in Figure 3 and Figure 4 that the aggregation does not have as great an impact as oversmoothing and that the activation function is responsible for the oversmoothing observed. This is a fallacy, the effect the authors observe is in fact the "dying ReLU" phenonmenon [1,2]. As they discuss, the Glorot intitialisation paired with ReLU is responsible and this is a known problem. A rigorous way to inspect the layed out hypothesis would be to ablate the activation function and initialization. Can we see the same effects when using other activation functions such as tanh, LeakyReLU, PReLU or ELU? Can we initialize in a way that mitigates this phenomenon? [7]
    - The proposed mitigation techniques have been known and used for years [3]. Recently, there have also been theoretical advancements as to why normalization and residual connections prevent oversmoothing [4-6]. So indeed, the proposed methods not only work well for the vanishing gradient problem, but also mitigate the oversmoothing caused by aggregation, leading to the absence of performance decline seen here. A rigorous way to approach this would be again to ablate only what is hypothesized to be respoinsible for the decline in performance - i.e. the vanishing gradient, but not oversmoothing. Normalization and residual connections do not work for this, as they prevent both problems.

- The presentation is suboptimal.
     - Most prominently the experiments in the form of the figures are not presented well. As an example take figure 4: The y-axis is not legible and necessary information is missing, e.g. which dataset was used, how many different random initializations were used, etc. Additionally, figures should be understandable by the figure caption alone. Al lot of details are missing from the caption. There is no reason not to plot the other two architectures from the previous ablation (those without the activation function).
    - The terms "oversmoothing" and "vanishing gradient" are not defined precisely. As a consequence, some statements can be confusing, e.g.: (Line 291) "Since the initial features are collapsed to zero in MLPs without proper normalization and residual connections, exponential oversmoothing is not a problem unique to GNNs." This statement directly contradicts the usual definition of oversmoothing as the phenomenon where "for GNNs with nondiverging weights, repeated message-passing invariably leads to the collapse of node signals into a one-dimensional subspace, regardless of initial features." [4]. This surely does not happen in conventional MLPs, where setting the weight matrices $W^{(i)} = I_n$ to the identity matrix displays no such degradation.

To conclude, the paper leads with an interesting proposition but fails to collect enough evidence to support the claims made. The paper does not add to the previous understanding of the vanishing gradient problem or oversmoothing or their magnitudes in the degradation of node features. The proposed mitigation techniques are known to work for both vanishing gradient and oversmoothing. Additionally, the presentation is subpar.

[1] Lu, Lu, et al. "Dying relu and initialization: Theory and numerical examples." arXiv preprint arXiv:1903.06733 (2019).

[2] Arroyo, Álvaro, et al. "On vanishing gradients, over-smoothing, and over-squashing in gnns: Bridging recurrent and graph learning." arXiv preprint arXiv:2502.10818 (2025).

[3] Chen, Ming, et al. "Simple and deep graph convolutional networks." International conference on machine learning. PMLR, 2020.

[4] Scholkemper, Michael, et al. "Residual Connections and Normalization Can Provably Prevent Oversmoothing in GNNs." The Thirteenth International Conference on Learning Representations.

[5] Kelesis, Dimitrios, Dimitris Fotakis, and Georgios Paliouras. "Analyzing the effect of residual connections to oversmoothing in graph neural networks." Machine Learning 114.8 (2025): 184.

[6] Chen, Ziang, et al. "Residual connections provably mitigate oversmoothing in graph neural networks." arXiv preprint arXiv:2501.00762 (2025).

[7] Kelesis, Dimitrios, Dimitris Fotakis, and Georgios Paliouras. "Reducing oversmoothing through informed weight initialization in graph neural networks" Applied Intelligence 55.7 (2025).

[8] Wu, Xinyi, et al. "Demystifying oversmoothing in attention-based graph neural networks." Advances in Neural Information Processing Systems 36 (2023): 35084-35106.

**Questions:**

- Can we see the same effects when using other activation functions such as tanh, LeakyReLU, PReLU or ELU?

- Can we initialize the weights in a way that mitigates this phenomenon?

- Could you clarify what you mean by "Since the initial features are collapsed to zero in MLPs without proper normalization and residual connections, exponential oversmoothing is not a problem unique to GNNs." (Line 291)?

- Can you provide more evidence that "Dying ReLU" is mistaken for oversmoothing by the broader GNN community other than Rusch et. al.? E.g. the oversmoothing analysis in [8] applies to any 1-Lipshitz function. They empirically evaluate using GeLU and find in their evaluation a similar result as this paper, in that, ReLU intensifies the problem.

---

> ### Author Response · Authors · 2025-12-03
>
> We thank the Reviewer for their careful evaluation and for recognizing (1) the importance of the question we raise, and (2) the depth of our review of prior literature. Below, the comments and questions are addressed in detail.
>
> > W1. The main hypothesis of the paper that "Oversmoothing is not a problem for the performance of deep GNNs" is not corroborated by the evidence layed out. Theoretical considerations are missing entirely.
>
> We respectively disagree that our study misses the theoretical consideration. We would like to emphasize that the claim in our paper aligns with existing theoretical insights, while we do highlight the points that have been mistakenly overlooked in the empirical validation of prior theories.
>
> According to the theory, oversmoothing occurs when the combined effects of aggregation and transformation push the oversmoothing measure in a reducing direction. In other words, if the transformation step offsets the effect of aggregation, oversmoothing does not occur.
>
> Building on this theoretical insight, we experimentally validated that oversmoothing does not occur in the presence of healthy gradients. We showed that prior empirical studies overlooked the fact that zero-collapsing driven by transformation and activation, which lead to vanishing gradient, dominates in deep GNNs rather than aggregation (Section 3.2). Moreover, our experiments (Figure 1 and Section 3.3) demonstrate that even when node features are initialized at a single point, the GCN can spread these features with healthy gradients, and performance improves as depth increases (Figure 8). This empirical result provides strong evidence that the effect of aggregation can be counteracted by transformation when optimization is effective.

---

> > ### Author Response · Authors · 2025-12-03
> >
> > > W2-1. The premise that researchers believe that oversmoothing is the single most impactful problem for deep GNNs is flawed. It makes little sense to believe that the problems plaguing MLPs would not carry over to GNNs (that use MLPs internally). However, oversmoothing is a new phenomenon and problem that is unique to GNNs, which may be the reason it has received the attention that is has. Additionally, oversmoothing has a much more devastating effect than e.g. vanishing gradients: For any sensible weights, the model will exponentially converge to an uniformative representation of the nodes and not be able to recover the initial signal. / Q3. Could you clarify what you mean by "Since the initial features are collapsed to zero in MLPs without proper normalization and residual connections, exponential oversmoothing is not a problem unique to GNNs." (Line 291)?
> >
> > We fully agree with the reviewer’s perspective that “it makes little sense to believe that the problems plaguing MLPs would not carry over to GNNs.” In fact, much prior work has overlooked this point, and clarifying it is precisely one of our goals.
> >
> > Conceptually, oversmoothing has been described as the phenomenon in which node representations become indistinguishable as layers increase. An attempt to provide a rigorous mathematical formulation was first made in [9], where oversmoothing is defined as the exponential decay of an oversmoothing measure across layers. Under this definition, even a vanilla MLP with ReLU activation and Glorot initialization exhibits oversmoothing. This occurs not because of aggregation, of course, but because of zero-collapsing, the feature elements converge to zero due to weights not being sufficiently far from zero in combination with the ReLU activation (and this is exactly what we meant in line 291).
> >
> > However, prior empirical studies on GNN's oversmoothing [8, 9, 10] interpreted the observed exponential decay as being caused by aggregation, without considering the dominant effect of zero-collapsing. This has reinforced the belief that oversmoothing is a GNN-specific phenomenon driven primarily by aggregation.
> >
> > Moreover, prior GNN work has largely overlooked the fact that GNNs inherit the optimization pathologies of MLPs, since GNN layers internally include MLPs. As a consequence, classical solutions commonly used in MLPs have not been sufficiently considered in GNN research. For example, zero-collapsing can be mitigated by He initialization, yet many GNN libraries and official implementations default to Glorot initialization (see lines 300–305). Similarly, most of oversmoothing-mitigation methods were isolately develped without sufficient consideration of skip connections and batch normalization, which are standard tools in MLPs to prevent vanishing gradients. As a result, as shown in Figure 7, many oversmoothing-mitigation methods can be outperformed simply by applying skip connections and batch normalization.
> >
> > Within this research landscape, the impact of oversmoothing has been overstated, reinforcing statements such as “oversmoothing has a much more devastating effect than vanishing gradients.” However, as implied by existing oversmoothing theory, oversmoothing does not occur when the effect of aggregation can be counteracted by the transformation step. Our experiments demonstrate this clearly: when skip connections and batch normalization are used to ensure healthy gradients, a GCN can take initial node features collapsed to a single point, spread them out, and even improve in performance with increasing depth (Figures 1 and 8).
> >
> > > W2-2. Figure 1 and Figure 5 depict the "recovery from oversmoothing" for GCN. However, there is no recovery: Initializing a GCN with all-ones does not start it off in a state of oversmoothing as it would for row-stochastic graph operators like the one used in GAT. Instead oversmoothing manifests in GCN as the exponential decay toward $\sqrt{d}$. So using $\sqrt{d}$ as the columns of $\mathbf{X}$ would actually start GCN in a state of oversmoothing. The GCN hence does not "recover" from oversmoothing in Figure 1 - it was never in a state of oversmoothing to begin with. The same idea leads to the phenomenon depicted in Figure 5. The claim that "Based on the empirical evidence, [...] GCNs are capable of escaping the oversmoothing regime." is not a consequence of the empirical evaluation shown.
> >
> > We acknowledge the distinction between the oversmoothing states in GAT and GCN, and this is precisely the point we aim to clarify in our paper (Section 3.1). However, even when initial node features are modified to a single point, the behavior of the GCN still demonstrates that the GCN can spread the features with healthy gradients, and performance improves as depth increases. In other words, the experiments in Section 3.3 (including Figure 1 and 5) effectively show that the effect of aggregation can be counteracted by transformation.

---

> > > ### Author Response · Authors · 2025-12-03
> > >
> > > > W3. The authors conclude from the experiments shown in Figure 3 and Figure 4 that the aggregation does not have as great an impact as oversmoothing and that the activation function is responsible for the oversmoothing observed. This is a fallacy, the effect the authors observe is in fact the "dying ReLU" phenonmenon [1,2]. As they discuss, the Glorot intitialisation paired with ReLU is responsible and this is a known problem. A rigorous way to inspect the layed out hypothesis would be to ablate the activation function and initialization. Can we see the same effects when using other activation functions such as tanh, LeakyReLU, PReLU or ELU? Can we initialize in a way that mitigates this phenomenon? [7]
> > > The proposed mitigation techniques have been known and used for years [3]. Recently, there have also been theoretical advancements as to why normalization and residual connections prevent oversmoothing [4-6]. So indeed, the proposed methods not only work well for the vanishing gradient problem, but also mitigate the oversmoothing caused by aggregation, leading to the absence of performance decline seen here. A rigorous way to approach this would be again to ablate only what is hypothesized to be respoinsible for the decline in performance - i.e. the vanishing gradient, but not oversmoothing. Normalization and residual connections do not work for this, as they prevent both problems. / Q1. Can we see the same effects when using other activation functions such as tanh, LeakyReLU, PReLU or ELU? /Q2. Can we initialize the weights in a way that mitigates this phenomenon?
> > >
> > > We believe there may be a misunderstanding regarding what our experiments in Figures 3 and 4 aim to demonstrate, and we would like to clarify this point.
> > >
> > > First, our results do not claim that “activation causes oversmoothing” or that the observed decay pattern is itself evidence of aggregation being unimportant. Rather, our purpose is to show that the commonly used oversmoothing measures in the literature cannot distinguish between aggregation-induced smoothing and zero-collapsing that arises from the combination of Glorot initialization and ReLU activation. In other words, what appears as “oversmoothing” in these measures can in fact be dominated by zero-collapsing, even when aggregation is absent. This observation is consistent with existing theoretical formulations of oversmoothing, which define it as exponential decay of a measure, not as a phenomenon exclusively caused by the graph aggregation operator.
> > >
> > > Second, for this reason, ablations over different activation functions (tanh, LeakyReLU, ELU, etc.) are not necessary for answering the scientific question we raise. Our goal is not to determine which activation function causes which pattern, but to show that the current oversmoothing metrics themselves conflate different mechanisms, making it impossible to attribute the measured decay solely to the aggregation operator. Changing the activation function would not address this conceptual issue, because the limitation lies in the metric, not in the specific activation used in the experiment.
> > >
> > > Third (Q2), regarding initialization: we already present results using He initialization (Section D.4), which is known to prevent zero-collapsing in MLP-type architectures. As shown in our experiments, applying He initialization largely eliminates the rapid decay that was previously attributed to oversmoothing. However, performance degradation still appears at greater depths, even after oversmoothing is significantly alleviated. This behavior aligns with our central claim, "the practical bottleneck for deep GNNs is not aggregation-induced oversmoothing, but vanishing gradients, which persist even when oversmoothing is mitigated."
> > >
> > > Therefore, our experiments using He initialization already address Q2: oversmoothing can be alleviated with appropriate initialization, yet performance degradation remains due to vanishing gradients.

---

> > > > ### Author Response · Authors · 2025-12-03
> > > >
> > > > > W4. The presentation is suboptimal. (Figure4의 y축 및 caption, oversmoothing과 vanishing gradient에 대한 정의)
> > > >
> > > > We appreciate the reviewer’s suggestion regarding the writing. We will incorporate this feedback into the revised manuscript.
> > > >
> > > > > Q4. Can you provide more evidence that "Dying ReLU" is mistaken for oversmoothing by the broader GNN community other than Rusch et. al.? E.g. the oversmoothing analysis in [8] applies to any 1-Lipshitz function. They empirically evaluate using GeLU and find in their evaluation a similar result as this paper, in that, ReLU intensifies the problem.
> > > >
> > > > We would like to clarify that the issue we raise is not limited to Rusch et al.[9]; similar misinterpretations appear in several influential works, including [8] and [10].
> > > >
> > > > In [8], the authors note that oversmoothing becomes more severe when ReLU is used, but nevertheless interpret the rapid exponential decay of the oversmoothing measure as evidence that oversmoothing is dominated by the combination of aggregation and activation. Likewise, [10] attributes the observed decay to aggregation and transformation. In both cases, however, the exponential decay arises primarily from zero-collapsing, yet it is interpreted as aggregation-induced oversmoothing. Thus, these works also conflate zero-collapsing with oversmoothing.
> > > >
> > > > This matters because these studies have been widely cited ([8] with 100+ citations, [9] with 400+, [10] with 1,000+) and have strongly contributed to the formation of the community’s belief that oversmoothing is the principal bottleneck preventing deep GNN construction. Although early theoretical work on oversmoothing had shown that node features asymptotically converge to one-dimensional subspace, its practical influence at depths of tens or hundreds of layers remained unclear. The theoretical results in [8] and [10] showing that oversmoothing occurs exponentially, and subsequent empirical validations in [8, 9, 10], where oversmoothing measures decay rapidly and reach to zero, were taken as strong evidence that oversmoothing is unavoidable and severe.
> > > >
> > > > However, when zero-collapsing is removed, for example, by using He initialization as shown in Section D.4, the oversmoothing measure still decays but very slowly, and remains well above zero even at depth 128. This indicates that the rapid decay observed in prior work is not inherent to aggregation, but primarily a consequence of zero-collapsing.
> > > >
> > > > For this reason, we believe that [8], [9], and [10] have unintentionally provided misleading empirical evidence that overestimates the practical impact of oversmoothing. Thus, we consider that clarifying the source of decay in these experiments is essential for accurately understanding the true role of oversmoothing in deep GNNs.
> > > >
> > > > [1] Lu, Lu, et al. "Dying relu and initialization: Theory and numerical examples." arXiv preprint arXiv:1903.06733 (2019).
> > > >
> > > > [2] Arroyo, Álvaro, et al. "On vanishing gradients, over-smoothing, and over-squashing in gnns: Bridging recurrent and graph learning." arXiv preprint arXiv:2502.10818 (2025).
> > > >
> > > > [3] Chen, Ming, et al. "Simple and deep graph convolutional networks." International conference on machine learning. PMLR, 2020.
> > > >
> > > > [4] Scholkemper, Michael, et al. "Residual Connections and Normalization Can Provably Prevent Oversmoothing in GNNs." The Thirteenth International Conference on Learning Representations.
> > > >
> > > > [5] Kelesis, Dimitrios, Dimitris Fotakis, and Georgios Paliouras. "Analyzing the effect of residual connections to oversmoothing in graph neural networks." Machine Learning 114.8 (2025): 184.
> > > >
> > > > [6] Chen, Ziang, et al. "Residual connections provably mitigate oversmoothing in graph neural networks." arXiv preprint arXiv:2501.00762 (2025).
> > > >
> > > > [7] Kelesis, Dimitrios, Dimitris Fotakis, and Georgios Paliouras. "Reducing oversmoothing through informed weight initialization in graph neural networks" Applied Intelligence 55.7 (2025).
> > > >
> > > > [8] Wu, Xinyi, et al. "Demystifying oversmoothing in attention-based graph neural networks." Advances in Neural Information Processing Systems 36 (2023): 35084-35106.
> > > >
> > > > [9] Rusch, T. Konstantin, Michael M. Bronstein, and Siddhartha Mishra. "A survey on oversmoothing in graph neural networks." arXiv preprint arXiv:2303.10993 (2023).
> > > >
> > > > [10] Kenta Oono and Taiji Suzuki. Graph neural networks exponentially lose expressive power for node classification. In International Conference on Learning Representations, 2020.

---

### Official Review · Reviewer_fTXG · 2025-10-30

**Soundness:** 3
**Presentation:** 3
**Contribution:** 2
**Rating:** 2
**Confidence:** 3

**Summary:**

In this paper, the authors study the problem of over-smoothing in Graph Neural Networks (GNNs). The authors argue that if we consider three of the main components of GNNs, aggregation, linear transformation, and non-linear activation, prior research studies have mistakenly confused over-smoothing with the vanishing gradient and zero-collapsing phenomenon, caused by transformation and activation rather than aggregation. The paper shows two propositions of over-smoothing which are specific to Graph Convolutional Networks (GCNs) and Graph Attention Networks (GATs): degree-scaled embedding convergence and uniform embedding convergence, implying that unless all degrees of the graph nodes are equal, or embeddings converge to the zero vector,

Their experiments show that aggregation does not play an important role in over-smoothing, while non-linear activation and linear transformation steps contribute to zero-collapsing, meaning that node embeddings converge toward zero.

**Strengths:**

- Interesting study showcasing that over-smoothing is sometimes confused with zero collapsing in related work
- Selecting the three base components of a GNN, and showing that aggregation does not play a significant role in over-smoothing

**Weaknesses:**

- The main weakness is the following: from my understanding when talking about GNNs it’s not just vanishing gradients. Over-smoothing and over-squashing can still appear with healthy gradients. Fixes like residual/identity mappings, careful init, normalization, JK connections, DropEdge, etc., help optimization and slow over-smoothing, but they don’t fully eliminate over-squashing or the diffusion-limit behavior. So the statement “performance degradation is not a phenomenon specific to GNNs, and it can be resolved by addressing the vanishing gradient problem” in line 417 is too strong. As seen in previous work by Arroyo et al. (2025), which is cited by the authors, over-squashing can be addressed by a combination of graph rewiring and vanishing gradient mitigation.
- The authors state in their contributions: "We show that the celebrated graph convolutional network can effectively overcome oversmoothing, provided that zero-collapsing is addressed separately". Again to my understanding the authors do not show or prove that in Section 4, rather show some results coming by specific datasets.

**Questions:**

- It would be interesting to connect all these phenomenon and under what properties/assumptions do they co-occur, e.g. vanishing gradients with oversmoothing

---

> ### Author Response · Authors · 2025-12-03
>
> We appreciate the Reviewer’s thoughtful assessment and their recognition that our work (1) clarifies conceptual confusion in prior studies, and (2) challenges prevailing assumptions regarding the role of aggregation in oversmoothing. We address the Reviewer’s comments and questions in detail below.
>
> > W1-1. The main weakness is the following: from my understanding when talking about GNNs it’s not just vanishing gradients. Over-smoothing and over-squashing can still appear with healthy gradients. Fixes like residual/identity mappings, careful init, normalization, JK connections, DropEdge, etc., help optimization and slow over-smoothing, but they don’t fully eliminate over-squashing or the diffusion-limit behavior. / Q1. It would be interesting to connect all these phenomenon and under what properties/assumptions do they co-occur, e.g. vanishing gradients with oversmoothing
>
> In our understanding, while aggregation-induced oversmoothing can indeed arise in the asymptotic regime, where the notion of healthy gradients is hard to be defined, this behavior does not persist at practical depths. In the regime of tens to a few hundred layers, whenever healthy gradients are maintained, the effect of aggregation-driven smoothing is counteracted by the transformation step and does not pose a practical limitation.
>
> This insight is consistent with existing oversmoothing theories. According to the theory, oversmoothing occurs when the combined effects of aggregation and transformation push the oversmoothing measure in a reducing direction. In other words, if the transformation step offsets the effect of aggregation, oversmoothing does not occur.
>
> In standard classification tasks, cross-entropy loss encourages features of the same class to be close together while pushing features of different classes apart. If the weight parameters are properly optimized, the transformation step will counteract the aggregation step, which drives all node features toward a one-dimensional subspace regardless of their class.
>
> Moreover, our experiments (Figure 1 and Section 3.3) demonstrate that even when initial node features are set to a single point, the GCN can spread the features with healthy gradients, and performance improves as the depth increases (Figure 8). This empirical result provides strong evidence that aggregation’s effect can be counteracted by transformation when optimization is effective.
>
> > W1-2. So the statement “performance degradation is not a phenomenon specific to GNNs, and it can be resolved by addressing the vanishing gradient problem” in line 417 is too strong. As seen in previous work by Arroyo et al. (2025), which is cited by the authors, over-squashing can be addressed by a combination of graph rewiring and vanishing gradient mitigation.
>
> While we agree that over-squashing is an important research topic in GNNs, we would like to clarify that it is outside the scope of our current work. However, even considering over-squashing, we still stand by our statement.
>
> We maintain that "performance degradation is not a phenomenon specific to GNNs", because performance degradation occurs even in MLPs due to the vanishing gradient.
>
> We also maintain that "it can be resolved by addressing the vanishing gradient problem". In our case, we show in Figure 3 that the primary cause of oversmoothing in deep GNNs is not aggregation, but rather zero-collapsing driven by transformation and activation, which leads to vanishing gradients. Our experiments in Section 4 demonstrate that when vanishing gradient problems are resolved by skip connections and batch normalization, performance degradation is alleviated, validating our claim.
>
> > W2. The authors state in their contributions: "We show that the celebrated graph convolutional network can effectively overcome oversmoothing, provided that zero-collapsing is addressed separately". Again to my understanding the authors do not show or prove that in Section 4, rather show some results coming by specific datasets.
>
> We would like to clarify that the contribution we mentioned specifically refers to the Section 3.3 (including Figure 1).The experiments in this section empirically support our insight that optimization prevents oversmoothing (i.e., addressing vanishing gradients rather than oversmoothing itself), which is aligned with previous theory (please see answer for W1-1). We believe it may be a misunderstanding to consider our findings as merely "some results coming by specific datasets".

---

### Official Review · Reviewer_78xB · 2025-10-31

**Soundness:** 3
**Presentation:** 3
**Contribution:** 2
**Rating:** 4
**Confidence:** 4

**Summary:**

This paper challenges the notion that oversmoothing is the main culprit for performance degradation with depth in GNNs, posits vanishing gradients as the root cause, and experimentally shows that skip connections and normalization can help overcome this.

**Strengths:**

1. The authors address, albiet not the first time, a common misconception regarding performance degradation in GNNs with depth, i.e. oversmoothing is not the sole culprit and vanishing gradients play a major role in practice. This is a pertinent problem to be highlighted.

2. The cause of confusion between the oversmoothing definitions for GCNs and GATs, and how that has affected commonly used metrics to measure oversmoothing is clarified.

**Weaknesses:**

1. There have been previous studies that identify similar reasons, primarily vanishing gradients and training problems, as a crucial factor in degraded performance with GNN depth rather than oversmoothing. [1,2,3,4]. Missing relevant literature should be discussed. In fact, some literature with the same insights as this paper are also already mentioned in the related work. This also challenges the novelty and contribution of the paper.

2. In section 3.2, the authors discuss initialization and zero collapsing. It is known that orthogonality can substantially reduce this if not prevent it. [3,4]. Such initializations are missing from the analysis.

[1] Decoupling the depth and scope of GNNs (NeurIPS 2021)
[2] Revisiting Oversmoothing in Deep GCNs (arXiv:2003.13663)
[3] Old can be Gold: Better Gradient Flow can make Vanilla GCNs Great (NeurIPS 2022)
[4] Are GATs Out of Balance? (NeurIPS 2023)

**Questions:**

1. Could the authors be more explicit about what is meant by 'when a model is properly trained' online 53? How would we define 'properly trained'?

2. What model/architecture is used for the cross-hatched bar in Fig 5?

3. For the experiment whre all node features are uniform, why are they set to the features of an arbitrary first node $X_{one}$ and not a random vector? How does the performance vary when nodes from different classes (or even different nodes from the same class) is used as $X_{one}$.?

4. Could the authors comment on other normalization techniques proposed for graphs such as nodeNorm, pairNorm GraphNorm etc, as opposed to batch normalization, and whether they are (or aren't) effective.

---

> ### Author Response · Authors · 2025-12-03
>
> We appreciate the Reviewer’s thoughtful assessment and recognition of (1) the importance of our research topic, and (2) clarification of the confusing definition of oversmoothing. We address the Reviewer’s questions and comments in detail below.
>
> > W1. There have been previous studies that identify similar reasons, primarily vanishing gradients and training problems, as a crucial factor in degraded performance with GNN depth rather than oversmoothing. [1,2,3,4]. Missing relevant literature should be discussed. In fact, some literature with the same insights as this paper are also already mentioned in the related work. This also challenges the novelty and contribution of the paper.
>
> We appreciate the reviewer for pointing out these relevant references. While we note that [1], in particular, attributes the accuracy drop in deep GNNs primarily to the aggregation process, which differs from the perspective we develop in this work, [2,3,4] indeed share a perspective similar to ours. However, we view these studies not as limiting our novelty, but as supporting evidence for our perspective. This is because **the goal of our paper is to argue that future progress in deep GNNs should shift away from oversmoothing-centric thinking and reconsider the true potential of deep GNNs**.
>
> We consider the novelty of our paper to arise from the following components that we carried out to achieve this goal:
>
> * (a) shedding light on prior works that have argued that oversmoothing is not the fundamental obstacl to deep GNN construction, or that have identified more classical causes as the primary issues (Section 5)
> * (b) identifying shortcomings in the existing empirical validations of oversmoothing theories, which have been regarded as strong evidence for the claim that oversmoothing is problematic (Section 3)
> * (c) demonstrating that using classical solutions is sufficient to resolve performance degradation, even outperforming various methods proposed to mitigate oversmoothing (Section 4.1)
> * (d) proposing a research direction for understanding when and why GNN performance can improve with depth, relating this behavior to dataset characteristics, beyond merely addressing performance degradation (Section 4.2)
>
> We will incorporate [2,3,4] into the revised manuscript to strengthen point (a).
>
> > W2. In section 3.2, the authors discuss initialization and zero collapsing. It is known that orthogonality can substantially reduce this if not prevent it. [3,4]. Such initializations are missing from the analysis.
>
> The purpose of Section 3.2 is not to propose solutions for zero-collapsing, but rather to clarify that prior empirical validations of oversmoothing may have mixed up oversmoothing with zero-collapsing. Therefore, we did not include an exhaustive analysis of all possible strategies that could mitigate zero-collapsing.
>
> However, we chose to include He initialization for a specific reason. We found that decaying rate of oversmoothing measure in prior studies is substantially reduced once He initialization is applied (Section D.5). We consider that this serves as additional strong evidence that oversmoothing research has overlooked classical solutions. Moreover, since He initialization is one of the most widely used initialization strategies, it may be difficult for readers to believe that previous studies consistently avoided using it. For this reason, we believe that discussing the effect of He initialization was important for ensuring the credibility and completeness of our analysis.
>
> By contrast, orthogonal or balanced initialization strategies such as those proposed in [3,4], while valuable, are not as standard or broadly adopted as He initialization and fall outside the scope of Section 3.2.
>
> > Q1. Could the authors be more explicit about what is meant by 'when a model is properly trained' online 53? How would we define 'properly trained'?
>
> By “properly trained,” we mean that the model is in a regime where the optimization process functions normally: the training loss decreases and converges, gradients remain sufficiently informative to update parameters, and training does not fail due to vanishing gradients or zero-collapsing. Our point is that GNNs can recover from oversmoothing once these basic optimization conditions are satisfied.
>
> > Q2. What model/architecture is used for the cross-hatched bar in Fig 5?
>
> The cross-hatched (blue) bar corresponds to the GCN with the original node features $\mathbf{X}$, whereas the solid blue bar uses the uniform features $\mathbf{X}_{\text{one}}$.

---

> > ### Author Response · Authors · 2025-12-03
> >
> > > Q3. For the experiment whre all node features are uniform, why are they set to the features of an arbitrary first node $\mathbf{X}_{\text{one}}$ and not a random vector? How does the performance vary when nodes from different classes (or even different nodes from the same class) is used as $\mathbf{X}_{\text{one}}$?
> >
> > We chose to set all node features to the feature of an arbitrary first node as one of several possible ways to modify all initial node features to a single point. The goal of this experiment is to show that GCN has the ability to spread features and perform well in the presence of healthy gradients, even when they are initialized at a single point. This conclusion holds regardless of whether the feature of the arbitrary node is a random vector, the feature of the first node, or the feature of any other node from the same class.
> >
> > > Q4. Could the authors comment on other normalization techniques proposed for graphs such as nodeNorm, pairNorm GraphNorm etc, as opposed to batch normalization, and whether they are (or aren't) effective.
> >
> > We focused on skip connections and batch normalization because our goal was to show that even simple, widely used classical techniques are sufficient to eliminate the performance degradation often attributed to deep GNNs and already outperform many oversmoothing-mitigation methods. A comprehensive evaluation of alternative normalization strategies such as NodeNorm, PairNorm, or GraphNorm is certainly interesting, but falls outside the scope of this work.
> >
> > [1] Decoupling the depth and scope of GNNs (NeurIPS 2021)
> >
> > [2] Revisiting Oversmoothing in Deep GCNs (arXiv:2003.13663)
> >
> > [3] Old can be Gold: Better Gradient Flow can make Vanilla GCNs Great (NeurIPS 2022)
> >
> > [4] Are GATs Out of Balance? (NeurIPS 2023)

---

### Official Review · Reviewer_M5NE · 2025-11-01

**Soundness:** 3
**Presentation:** 3
**Contribution:** 2
**Rating:** 4
**Confidence:** 4

**Summary:**

The authors argue that aggregation alone has a minimal impact on oversmoothing, whereas the nonlinear activation and linear transformation steps are responsible for oversmoothing and 'zero-collapsing phenomenon'. Some techniques such as residual connections and batch normalization can be used to mitigate zero-collapsing. The main contributions are: 1. the authors clarify the previous misunderstanding of zero-collapsing as oversmoothing 2. the authors clarify that the aggregation step in GNNs has a marginal effect on oversmoothing compared to transformation and activation. 3. the authors empirically show residual connections and batch normalization are efficient solutions to mitigate oversmoothing. The paper conducts a decomposition of the three GNN operations — aggregation, linear transformation, and nonlinear activation — and quantifies their combined contributions to oversmoothing. Empirical results show performance improvement in deep GNN on skip connection and batch normalization.

**Strengths:**

1.The paper challenges a previous core assumption in GNN research that oversmoothing has been overstated, whereas vanishing gradient is the main issue caused by transformation and activation rather than aggregation.
2. The author did a systematic component analysis by isolating the effects of aggregation, transformation, and activation, and clearly show where oversmoothing actually arises with different combination of components.
3. The experimental results demonstrate that deep GNNs with batch normalization and skip connections can perform effectively without oversmoothing.

**Weaknesses:**

1.Although the author empirically show the effects of aggregation, transformation, and activation on oversmoothing, the paper provides limited theoretical proofs explaining why aggregation has marginal impact.
2. Skip connections and normalization techniques are wide-known solutions to oversmoothing. While the findings encourage deeper models, the paper provides limited guidance on how to practically design or train them beyond skip connections and normalization.

**Questions:**

Can the author show individual impacts of transformation and activation on oversmoothing? Besides skip connections and normalization techniques, can the authors provide additional solutions directly target activation and transformation?

---

> ### Author Response · Authors · 2025-12-03
>
> We sincerely thank the Reviewer for their thoughtful evaluation and for highlighting that our work (1) challenges the prevailing assumption in GNN research, (2) provides a systematic component-wise analysis, and (3) demonstrates the effectiveness of classical solutions in deep GNNs. Below, we address the Reviewer’s comments and questions in detail.
>
> > W1. Although the author empirically show the effects of aggregation, transformation, and activation on oversmoothing, the paper provides limited theoretical proofs explaining why aggregation has marginal impact.
>
> We would like to emphasize that empirical results in our paper align with existing theories, and therefore do not require additional theoretical proofs.
>
> To clarify this alignment, let us provide more specific details on how our results correspond to previous experiments. The existing theories in [1] and [2], for example,  present the upper bounds on the oversmoothing measure after applying aggregation, transformation, or activation, in terms of the oversmoothing measure before each operation (See equation A3 and A5). [3] focuses on analyzing how the oversmoothing measure $\mu_{\text{wu}}$ evolves under the combination of aggregation and activation. However, both [1] and [3] have verified their claims using GNNs with all components ($\sigma$ + A + T), which correspond to the blue curves in Figure 3.
>
> In contrast, we decompose the GNN layer into its aggregation, transformation, and activation components to isolate the individual influence of each step. The effect of applying only aggregation can be observed in the red curves in Figure 3. These red curves show that aggregation alone induces a decay of specific measures, $\mu_{\text{cai}}$ for GCN and $\mu_{\text{rusch}}$ for GAT, although the decay occurs at a much slower rate compared to the blue curves.
>
> These findings are fully consistent with existing theory: they do not contradict prior analyses and remain compatible with them. Instead, our results reveal that earlier empirical validations have tended to underestimate the influence of transformation and activation, and to overestimate the influence of aggregation.
>
> > W2. Skip connections and normalization techniques are wide-known solutions to oversmoothing. While the findings encourage deeper models, the paper provides limited guidance on how to practically design or train them beyond skip connections and normalization.
>
> We agree with the reviewer that our paper does not provide additional architectural guidelines for deep GNNs, beyond the use of skip connections and batch normalization.
>
> **Our contribution is to provide a new research perspective rather than introduce new practical design suggestions**. We will make this distinction clearer in the revised manuscript.
> Our experiments using skip connections and batch normalization show that:
> * performance degradation, which is often attributed to oversmoothing in deep GNNs, can already be resolved through these simple, classic solutions (Figure 6), and
> * these classic solutions outperform a variety of prior methods specifically proposed to mitigate oversmoothing and enable deep GNN construction (Figure 7).
>
> These findings indicate that prior research may have overly centered on oversmoothing while overlooking more classic deep-learning challenges such as vanishing gradients.
>
> Therefore, we argue that future progress in deep GNNs should shift away from oversmoothing-centric thinking and reconsider the true potential of deep GNNs. We believe that this reframing provides a meaningful and valuable contribution, even without providing detailed architectural prescriptions.
>
> > Q1-1. Can the author show individual impacts of transformation and activation on oversmoothing?
>
> We have added the individual effects of transformation and activation in Figure 3 of the revised manuscript. When applied alone, the transformation step keeps the oversmoothing measures roughly constant, and the activation step yields a precisely constant pattern. These observations further support our main finding that the rapid decay observed in Figure 3 is not caused by either transformation or activation in isolation, but instead arises from their combined effect, which induces the zero-collapsing phenomenon as discussed in the manuscript.
>
> > Q1-2. Besides skip connections and normalization techniques, can the authors provide additional solutions directly target activation and transformation?
>
> A direct way to mitigate zero-collapsing caused by the transformation and activation steps is to improve weight initialization. As described in lines 296–307 of the manuscript, zero-collapsing is greatly alleviated simply by replacing Glorot initialization with He initialization. However, we note that He initialization alone does not completely resolve the vanishing gradient problem, which requires skip connection and normalization.

---

> > ### Author Response · Authors · 2025-12-03
> >
> > [1] Chen Cai and Yusu Wang. A note on over-smoothing for graph neural networks. In ICML
> > Graph Representation Learning and Beyond (GRL+) Workshop, 2020.
> >
> > [2] Kenta Oono and Taiji Suzuki. Graph neural networks exponentially lose expressive power for node classification. In International Conference on Learning Representations, 2020.
> >
> > [3] Xinyi Wu, Amir Ajorlou, Zihui Wu, and Ali Jadbabaie. Demystifying oversmoothing in attention-based graph neural networks. In Thirty-seventh Conference on Neural Information Processing Systems, 2023.

---

### Meta-Review · Area_Chair_dhjV · 2026-01-05

**Summary:**

Although the paper touches on an important question, namely the precise role of oversmoothing in deep GNNs, all reviewers initially recommended rejection of the manuscript. The primary concerns were that (i) despite making exceptionally strong claims, the paper does not provide sufficient theoretical justification or convincing empirical evidence; (ii) it contains logical inconsistencies and draws potentially incorrect conclusions; and (iii) it does not offer substantive new insights into the vanishing gradient problem or oversmoothing beyond what is already well understood in the literature.

**Reviewer Concerns:**

Minor reviewer concerns were successfully addressed and clarified during the rebuttal. However, the major points of criticism remain insufficiently resolved. In particular, the paper does not provide novel theoretical insights. The impact of vanishing gradients on deep GNNs has been well understood for many years, and the observation that residual connections facilitate training deeper GNNs is likewise unsurprising, given the extensive prior theoretical and empirical work in this direction. Taking all of these points into consideration, the paper does not meet the bar for acceptance at ICLR, even after the rebuttal.

**Reviewer Scores:**

I do not believe the reviewers would have raised their scores as the major points of criticism were not convincingly addressed by the rebuttal.

---

### Decision · Program_Chairs · 2026-01-26

Reject